# The dopamine transporter antiports potassium to increase the uptake of dopamine

Solveig G. Schmidt [1], Mette Galsgaard Malle [2,3,5], Anne Kathrine Nielsen[1,4,5], Søren S.-R. Bohr [2,3], Ciara F. Pugh[1], Jeppe C. Nielsen [1], Ida H. Poulsen[1], Kasper D. Rand [4], Nikos S. Hatzakis[2,3] & Claus J. Loland [1✉]

The dopamine transporter facilitates dopamine reuptake from the extracellular space to terminate neurotransmission. The transporter belongs to the neurotransmitter:sodium symporter family, which includes transporters for serotonin, norepinephrine, and GABA that utilize the $Na^+$ gradient to drive the uptake of substrate. Decades ago, it was shown that the serotonin transporter also antiports $K^+$, but investigations of $K^+$-coupled transport in other neurotransmitter:sodium symporters have been inconclusive. Here, we show that ligand binding to the *Drosophila*- and human dopamine transporters are inhibited by $K^+$, and the conformational dynamics of the *Drosophila* dopamine transporter in $K^+$ are divergent from the apo- and $Na^+$-states. Furthermore, we find that $K^+$ increases dopamine uptake by the *Drosophila* dopamine transporter in liposomes, and visualize $Na^+$ and $K^+$ fluxes in single proteoliposomes using fluorescent ion indicators. Our results expand on the fundamentals of dopamine transport and prompt a reevaluation of the impact of $K^+$ on other transporters in this pharmacologically important family.

[1] Laboratory for Membrane Protein Dynamics, Department of Neuroscience, Faculty of Health and Medical Sciences, University of Copenhagen, Copenhagen, Denmark. [2] Nano-Science Center, Department of Chemistry, Faculty of Science, University of Copenhagen, Copenhagen, Denmark. [3] Novo Nordisk Foundation Center for Protein Research, Faculty of Health and Medical Sciences, University of Copenhagen, Copenhagen, Denmark. [4] Protein Analysis Group, Department of Pharmacy, Faculty of Health and Medical Sciences, University of Copenhagen, Copenhagen, Denmark. [5] These authors contributed equally: Mette Galsgaard Malle, Anne Kathrine Nielsen. ✉email: cllo@sund.ku.dk

Dopaminergic signaling is involved in higher-order brain functions such as movement, mood, motivation, and learning[1,2]. The dopamine transporter (DAT) is responsible for the clearance of dopamine from the extracellular space and hence plays an important role in dopamine signaling intensity[2] and in refilling stores of dopamine[3]. DAT malfunction has been associated with parkinsonism and ADHD[4,5]. In addition, DAT is the molecular target for pharmaceutical drugs such as methylphenidate, bupropion, and modafinil[3], and also the rewarding and addictive effects of psychostimulants are linked to their interaction with the transporter[6–8]. The compounds that target DAT are classified into inhibitors and substrates but the underlying differences in mechanisms of drug action are generally poorly understood, including the reverse transport mode induced by amphetamine[9]. A prerequisite for investigating the mechanisms of action of the drugs is to understand the basic transport mechanism of the transporter.

DAT belongs to the neurotransmitter:sodium symporter (NSS) family[10], which also includes transporters of serotonin, norepinephrine, GABA, metabolites, and amino acids. NSSs are found in species from eukaryotes, bacteria, and archaea. The NSSs share a common structural fold consisting of 12 transmembrane segments (TMs) connected by extracellular- (EL) and intracellular (IL) loops[11]. Transport is proposed to occur by a rocking-bundle mechanism, where the central binding site for substrate and ions is alternately accessible from either side of the cell membrane, caused by the movement of TM 1, 2, 6, and 7, relative to TMs 3, 4, 8, and 9[12–14].

In the brain, the transporters for dopamine, norepinephrine (NET) and serotonin (SERT), are localized extrasynaptically in monoaminergic neurons[15], which entails that during resting potential the transporters will be exposed to both an inward directed $Na^+$ gradient and an outward directed $K^+$ gradient. It has been long known that SERT in addition to the symport of $Na^+$ catalyzes antiport of $K^+$ [16]. The $K^+$ antiport is permissive and increases the $k_{cat}$ for serotonin[17], but the affinity for $K^+$ has not been determined. This feature is thought to be unique for SERT among the NSSs[18]. However, we have observed an effect of $K^+$ on the transport properties and conformational state of LeuT, a prokaryote NSS[19–21], with an apparent affinity for $K^+$ to LeuT around 175 mM. Although still controversial[22], it opens for the possibility that $K^+$ counter-transport might be an evolutionary

preserved mechanism within the NSS family that previously have been overlooked.

Here, we find that, analogue to previous observations from SERT, both the human and *Drosophila* DAT variants are also affected by manipulating the $K^+$ concentration. We see that $Na^+$-dependent ligand binding is dose-dependently decreased by $K^+$ and that $K^+$ induces a DAT conformation, which is significantly different from both the apo- and the $Na^+$-bound form. Intra-vesicular $K^+$ markedly increased [³H]dopamine transport when purified *Drosophila* DAT (dDAT) was reconstituted into proteoliposomes (PLs). Loading the PLs with a fluorescent $K^+$ indicator and following the fluorescence in a single vesicle setup, we observed an accelerated fluorescence decrease when applying dopamine, suggesting a $K^+$ antiport.

## Results

**$K^+$ shows competitive binding relative to $Na^+$.** To address the role of $K^+$ in the NSS mechanism, we turned to dDAT, which shares more than 50% sequence identity with the mammalian dopamine transporters[23–25]. At this stage, it has not been possible to purify the human DAT (hDAT) in sufficient quantities and stability needed for the required experiments. However, dopamine affinity and transport rate by dDAT is comparable to hDAT, and the pharmacological profile of dDAT lies between that of hDAT and the human NET[25], making dDAT a suitable model for studies of NSS transport. Using dDAT, we investigated whether $K^+$ is a component in dopamine transport.

The potent NET inhibitor nisoxetine binds dDAT with high affinity (Supplementary Fig. 1a)[25]. We investigated if there are indications that $K^+$ can bind to dDAT by observing how [³H]nisoxetine binding was influenced by $K^+$. In SERT and LeuT the interaction with $K^+$ excludes substrate binding and is competitive to $Na^+$ [19,26]. If a $K^+$ binding site exists in dDAT, it could have similar mechanistic features. To address this, we expressed full-length dDAT-8-His protein in suspension HEK293 cells (Expi293F) using the BacMam system[27,28] and purified the transporter[23]. We probed the $Na^+$-dependence of [³H]nisoxetine binding to dDAT in mixed detergent-lipid micelles by the scintillation proximity assay[29] (Fig. 1a). We found that $Na^+$ supports the binding of 120 nM [³H]nisoxetine with an $EC_{50}$ of 60 [53;67] mM (mean [SEM interval]), whereas $K^+$ did not

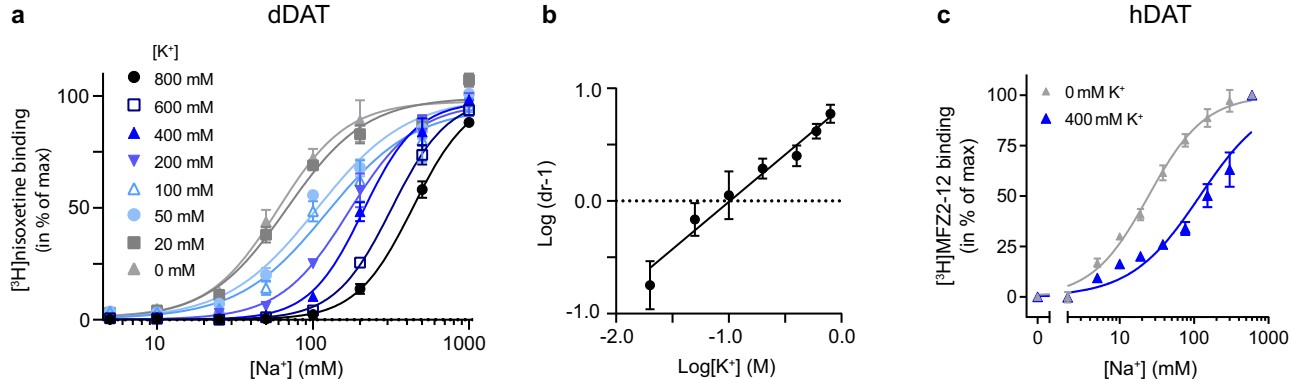

**Fig. 1 $Na^+$-dependent ligand binding to DAT is inhibited by $K^+$. a** $Na^+$-dependent [³H]nisoxetine binding to purified dDAT in mixed micelles in increasing $K^+$ concentrations ($n = 3$ independent experiments performed in triplicates). Lines are fit to dose-response functions (Supplementary Table 1). **b** Schild plot derived from $EC_{50}$ values in (**a**). The fit is a linear-regression with a slope of 0.84 [0.75, 0.93] (mean [SEM interval]). The slope is determined not to be significantly different from 1 (sum-of-squares F-test, $P = 0.0872$), which indicates that $K^+$ acts as a competitive inhibitor to $Na^+$ binding. The derived binding constant ($K_B$) for $K^+$ binding was 102 [100;105] mM (mean [SEM interval]). **c** $Na^+$-dependent binding of the cocaine analogue [³H]MFZ 2-12 to membranes prepared from Expi293F cells expressing hDAT. Addition of 400 mM $K^+$ increased the $EC_{50}$ value from 24 [23;26] to 121 [106;134] mM (mean [SEM interval], $n = 3$ independent experiments performed in triplicates). Data points in (**a**, **b**, **c**) represent means ± SEM (error bars). Data are provided as a Source Data file.

support [3H]nisoxetine binding (Supplementary Fig. 1b). We then included increasing $K^+$ concentrations and found that $K^+$ dose-dependently decreased the $EC_{50}$ value for $Na^+$-dependent [3H]nisoxetine binding (Fig. 1a and Supplementary Table 1). A Schild plot analysis[30] of the change in $EC_{50}$ values as a function of added $K^+$, estimated the affinity ($K_B$) for $K^+$ to 102 [100;105] mM (mean [SEM interval]). The slope of the linear regression in the plot was not significantly different from 1 suggesting that $K^+$ acts as a competitive inhibitor of $Na^+$ binding in dDAT (Fig. 1b). We next assessed whether the inhibitory effect of $K^+$ is shared between dDAT and hDAT. We expressed hDAT in Expi293F cells, harvested the membranes, and measured the $Na^+$-dependence of binding of the high affinity, cocaine analog [3H]MFZ 2-12[31] to hDAT in the presence and absence of $K^+$ (Fig. 1c). The $EC_{50}$ value for $Na^+$ binding was 24 [23;26] mM (mean [SEM interval]), but increased to 121 [106;134] mM (mean [SEM interval]) by the addition of 400 mM $K^+$, indicating a similar inhibitory effect of $K^+$ in dDAT and hDAT.

**$K^+$ induces specific structural dynamics in dDAT**. If dDAT possesses a $K^+$ binding site with implication for function, then $K^+$ binding would be expected to impact the dynamics of the dDAT structure. To investigate this, we probed the influence of $K^+$ on the overall conformational dynamics of dDAT by hydrogen-deuterium exchange mass spectrometry (HDX-MS). The rate of hydrogen-deuterium exchange (HDX) of backbone amide hydrogens correlates with the specific hydrogen bonding of higher-order protein structure[23,32]. For instance, if the binding of an ion stabilizes the region of a protein, it will typically be reflected in the HDX profile of that region. Accordingly, we probed the solution-phase HDX profile of purified dDAT in mixed micelles in 200 mM KCl. The time-dependent HDX was monitored for 85 identified dDAT peptides[23] covering ~77% of the protein sequence (Fig. 2, Supplementary Table 2 and 3), thus providing us with a comprehensive view of the structural dynamics of the protein (Supplementary Fig. 2).

We compared the HDX data of dDAT in the presence of 200 mM $K^+$ relative to a control buffer of similar ionic strength where $K^+$ was substituted with $Cs^+$. $Cs^+$ was chosen as inert ion over $NMDG^+$ because it is more $K^+$ and $Na^+$-like in size and because it enables comparison to previous HDX-MS studies of dDAT[23]. We found that many regions of dDAT underwent decreased HDX in the $K^+$-containing buffer, which indicated widespread stabilization of dDAT structural dynamics by $K^+$ relative to $Cs^+$ (Fig. 2 and Supplementary Fig. 2). The stabilizing effects were localized to the core domain (TM1, 2, 6, and 7) and minor parts of TM10, 12, and the C-helix, as well as the EL1, 2, 3, 4, and IL3, 4, 5. The $K^+$ stabilizing effect was mainly located to regions suggested to be involved in the substrate transport process as also observed previously for $Na^+$ in dDAT[23]. This $K^+$-induced stabilization of distinct, functionally relevant regions of dDAT indicates a specific $K^+$ binding site in the transporter.

The comparison of the HDX profile of dDAT in $Na^+$- versus the $K^+$ state revealed that even though both ions impact the conformational dynamics in many of the same regions, the differences in HDX ($\Delta$HDX) are of opposing magnitude for several regions (Fig. 3 and Supplementary Fig. 2). This argues against that $K^+$ simply substitutes for $Na^+$ and induces a $Na^+$-bound conformation. Relative to $Na^+$, $K^+$ induced structural stabilization of TM1b, part of the hinge region of TM1, parts of TM7, and in loop regions on the extracellular side in EL2, 3, 4 and 6. In contrast, most intracellular regions were destabilized in $K^+$ compared to the $Na^+$ state. Specifically, TM1a, TM6b, IL3, the intracellular part of TM7, and IL4 along with the intracellular

parts of TM8 and TM9, showed increased dynamics in the $K^+$ state. Increased dynamics were also observed on the extracellular face of the transporter in EL2, part of EL3, and TM6a. The higher degree of stabilization on the extracellular face and intracellular destabilization, could suggest that $K^+$ induces a more inward facing dDAT state than $Na^+$. The destabilization of the intracellular regions TM1a, IL 3 and IL 4 are consistent with an inward open conformation where TM1a flips away from the core into the lipid bilayer, and the intracellular gate is destabilized to allow access to the substrate binding site from the intracellular side. In good agreement, this is the conformation SERT has been suggested to adopt in its $K^+$ bound state[33]. Furthermore, HDX-MS analysis of SERT has revealed a destabilization in TM1a upon binding of $K^+$ [34], which also align with our findings in dDAT.

**dDAT transport of dopamine into proteoliposomes is increased by $K^+$**. To explore the functional relevance of $K^+$ to dopamine transport, we reconstituted purified dDAT into liposomes with tight control of the intra-vesicular ionic content. The reconstituted dDAT transported [3H]dopamine into the lumen of the PLs in the presence of an inward-directed $Na^+$ gradient (Fig. 4a). No dopamine uptake was observed in the presence of the inhibitor nortriptyline, or when the $Na^+$ gradient was dissipated (Supplementary Fig. 3a), which indicated that the observed uptake was dDAT mediated and not a measure of dopamine binding to dDAT.

We then compared [3H]dopamine transport activity from vesicles containing intra-vesicular $K^+$ to vesicles containing either $Cs^+$ or $NMDG^+$. At all times, chloride was used as the corresponding anion. Uptake data were correlated to the relative amount of active dDAT in the reconstituted system (Supplementary Fig. 3b). The inwardly directed $Na^+$ gradient could drive dopamine uptake regardless of the intra-vesicular cation, but the concentrative capacity decreased to about 20% when $K^+$ was substituted with either $Cs^+$ or $NMDG^+$ (Fig. 4b and Supplementary Table 4). The level of dopamine uptake in $Cs^+$ and $NMDG^+$ was comparable, which indicates that neither of the ions affects the transport. The transport kinetics were further investigated by [3H]dopamine saturation uptake experiments (Fig. 4c and Supplementary Table 5). The $K_m$ for [3H]dopamine was comparable to previous observations of dDAT in intact cells[35]. Based on binding data from re-solubilized PLs, we estimated that approximately 15% of the dDAT retained activity after reconstitution. In PLs containing 200 mM $K^+$ $V_{max}$ was on average 1.53 pmol min$^{-1}$ and the $k_{cat}$ was estimated to 1.1 min$^{-1}$ (Supplementary Table 6). However, we observed a significant decrease in transport velocity, when the intra-vesicular $K^+$ concentration was lowered from 200 mM to 150 and 100 mM, suggesting that the $V_{max}$ of dopamine transport is dependent on intra-vesicular $K^+$. We also observed a drop in transport velocity when we dissipated the $K^+$ gradient, by having 100 mM $K^+$ on both sides (Fig. 4d), suggesting that the $K^+$ gradient contributes with a driving force for dopamine transport.

**Dopamine transport into cells is inhibited by $K^+$ in the uptake buffer**. To investigate how $K^+$ affects dopamine uptake in a more complex system, we expressed hDAT in COS-7 cells. We compared the uptake of [3H]dopamine over 7.5 min in COS-7 cells in a buffer with 30 mM $Na^+$ and either 105 mM $K^+$ or $NMDG^+$. We observed a significant decrease in $V_{max}$ and an unaltered $K_m$ for [3H]dopamine uptake when applying the external $K^+$ (Fig. 4e). Although the experiment could influence many cellular processes, the effect is similar to our findings for [3H]dopamine uptake in PLs containing dDAT and could be interpreted as an inhibitory effect caused by the abolishment of the $K^+$ gradient.

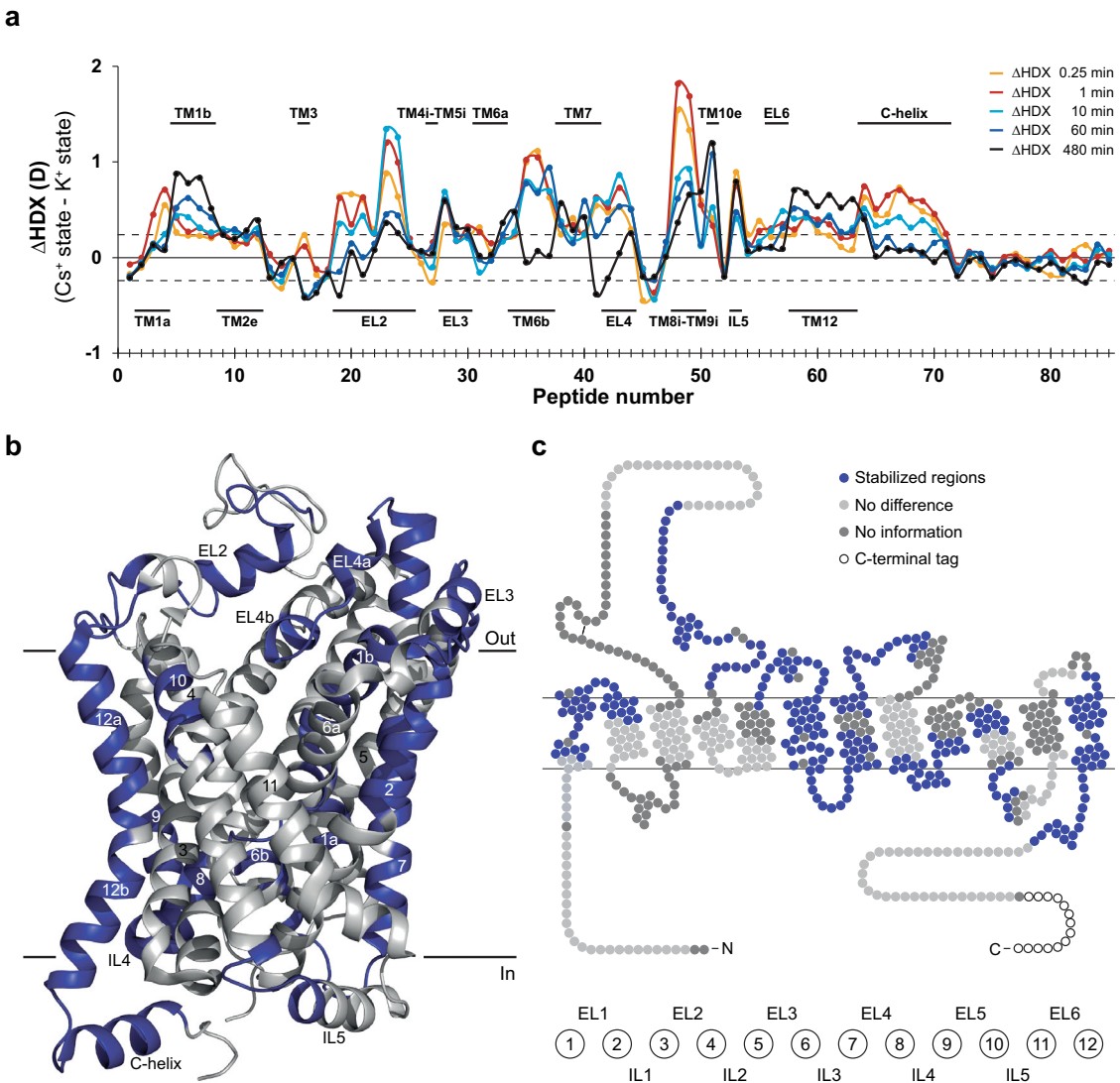

**Fig. 2 K⁺ impacts the conformational dynamics of dDAT. a** Differential uptake chart comparing differences in the average deuterium uptake (ΔHDX in deuterons (D)) between the apo (Cs⁺) state to the K⁺ state for the 85 identified peptides at the five sampled time points (orange: 0.25 min; red: 1 min; cyan: 10 min; blue: 1 h; black: 8 h). The individual dDAT peptides are arranged along the x-axis starting from the N-terminal and ending on the C-terminal. The peptide number refers to Supplementary Data Table 3. Positive and negative values indicate reduced and increased HDX, respectively, for the K⁺-containing sample. Structural dDAT motifs are marked along the x-axis. **b, c** Data from the differential uptake chart exhibiting significant differences (Student's *t*-test *p*-value < 0.01) in deuterium uptake between the apo (Cs⁺) and the K⁺ states mapped onto regions of dDAT on cartoons based on (**b**) the crystal structure (PDB ID: 4XP1) and (**c**) a snake diagram. Unchanged and uncovered regions are colored light- and dark grey, respectively. Values represent means (*n* = 3 independent measurements). The dotted lines (± 0.24 D) mark a threshold value for the 95% confidence interval, calculated from the pooled SD for all time points. Data are provided as a Source Data file.

Similarly, we expressed dDAT in COS-7 cells and performed the same uptake experiments. Again, we observed a marked decrease in dopamine uptake, in the presence of extracellular K⁺ although the lower affinity for dopamine to dDAT did not allow for a precise estimate of $V_{max}$ and $K_m$ (Fig. 4f). Taken together the uptake from whole cells could indicate that a dissipation of the K⁺ gradient not only affects transport in PLs but also in cells and that the effect is also observed for hDAT.

**Time-resolved flux of Na⁺ and K⁺ mediated by dDAT.** To obtain a more direct measure of dDAT mediated ion flux, we followed in real-time, the change in intra-vesicular ion concentration in a single vesicle setup. The PLs were tethered on a surface by biotinylated lipids and visualized with ATTO-665 membrane dye using total internal reflection fluorescence

microscopy (TIRFm)[36–38]. To examine the ion flux we encapsulated either a fluorescent Na⁺ indicator (Sodium Green™, Tetra (Tetramethylammonium) Salt) or a fluorescent K⁺ indicator (ION Potassium Green-2 TMA+ salt) in the PLs (Supplementary Fig. 4a–c). The PLs were passivated on a glass surface at the bottom of a flow cell. This allowed us to measure the relative changes in ion concentration inside individual PLs as a function of the intensity change of the fluorescence signal from the indicators (Fig. 5a). The procedure maintains the spherical morphology and structural integrity of the liposomes[36,37]. Comparison between dynamic light scatter (DLS) and TIRFm measurements revealed an expected log normal size distribution of the PLs centered at ~181 nm with no change upon indicator encapsulation (Supplementary Fig. 4d, e). Because PLs are below or close to the optical resolution, they will appear as diffraction limited spots in the TIRFm recordings. The 200 nm penetration depth ensures full

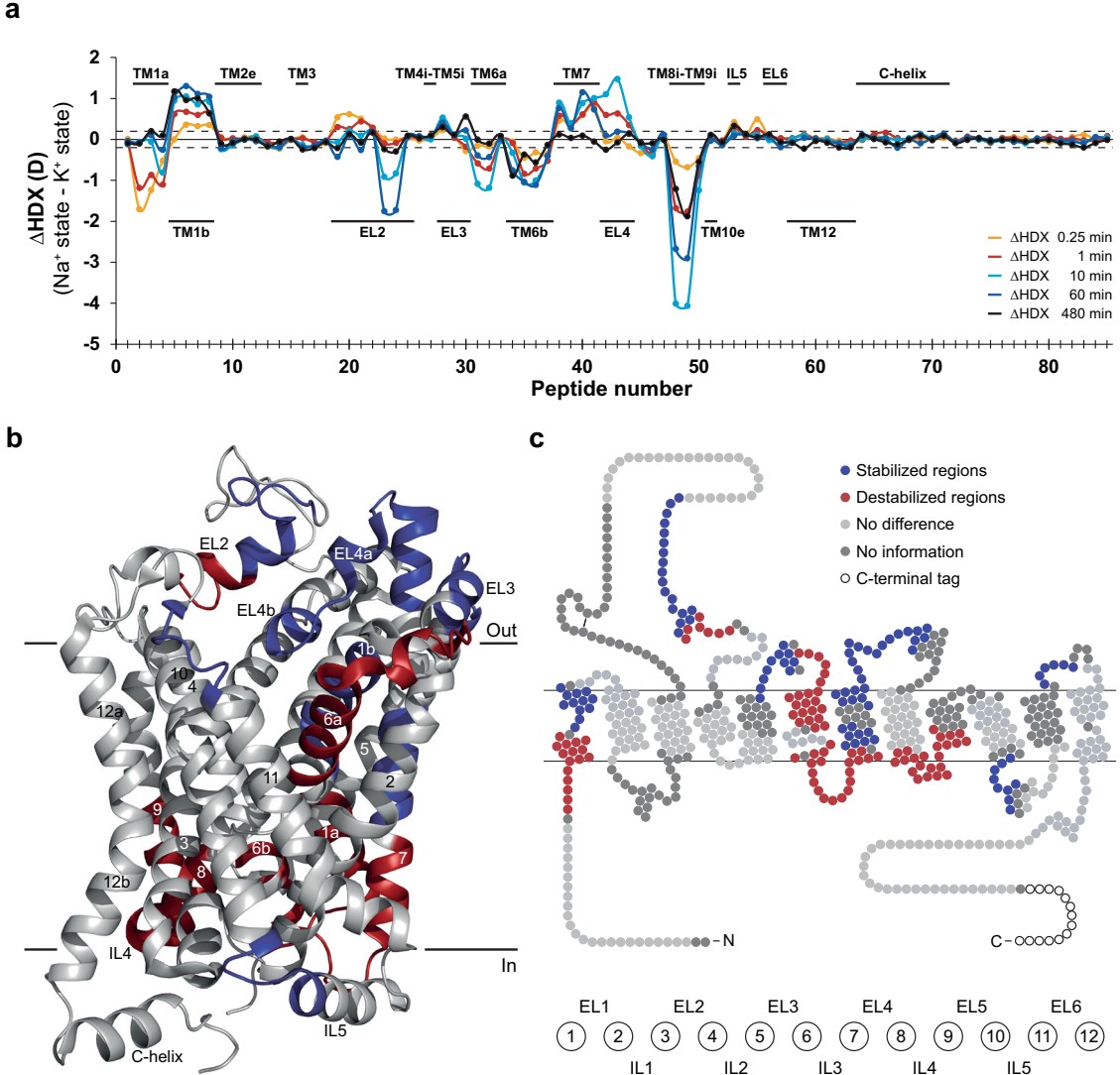

**Fig. 3 K$^+$ affects dDAT conformational dynamics differently than Na$^+$. a** Differential uptake chart comparing differences in the average deuterium uptake (ΔHDX in deuterons (D)) between the Na$^+$ state and the K$^+$ state for the 85 identified peptides at the five sampled time points (orange: 0.25 min; red: 1 min; cyan: 10 min; blue: 1 h; black: 8 h). The individual dDAT peptides are arranged along the x-axis starting from the N-terminal and ending on the C-terminal. The peptide number refers to Supplementary Data Table 3. Positive and negative values indicate reduced and increased HDX, respectively, for the K$^+$-containing sample. Structural dDAT motifs are marked along the x-axis. **b, c** Data from the differential uptake chart exhibiting significant differences (Student's t-test p-value < 0.01) in deuterium uptake between the Na$^+$ and the K$^+$ states mapped onto regions of dDAT on cartoons based on (**b**) the crystal structure (PDB ID: 4XP1) and (**c**) a snake diagram. Unchanged and uncovered regions are colored light- and dark grey, respectively. Values represent means (n = 3 (K$^+$ state) and 6 (Na$^+$ state) independent measurements). The dotted lines (± 0.20 D) mark a threshold value for the 95% confidence interval, calculated from the pooled SD for all time points. Data are provided as a Source Data file.

illumination and unbiased analysis of the PLs as we have shown in the past[36–40]. Additionally, uptake of [³H]dopamine was maintained (Supplementary Fig. 5a, b).

We evaluated how PLs containing the Na$^+$ indicator responded to the application of external Na$^+$ and dopamine (Fig. 5b). About half of the detected vesicles responded to the application of Na$^+$ and dopamine with an exponential increase in fluorescence intensity from the indicator followed by a plateau (Fig. 5b, c), indicating an expected increase in intra-vesicular Na$^+$ concentration during dopamine uptake. The number of responding vesicles decreased to about half when only Na$^+$ was added into the flow cell, indicating that a fraction of the PLs would take up Na$^+$ even in the absence of dopamine. Nortriptyline (500 µM) blocked Na$^+$ uptake within the timeframe of the experiment, confirming that our observations are dDAT-mediated. We did

not detect passive Na$^+$ leakage through the lipid bilayer (Fig. 5c). After reaching the plateau, the fluorescence intensity of the Na$^+$ indicator decreased. We consider this an inherent property of the indicator associated with inner filter effect (Supplementary Fig. 6) disconnected from the Na$^+$ influx, and was therefore not included in the further calculations.

The measurement of dDAT-mediated Na$^+$ flux into single PLs, lead us to proceed with the K$^+$ indicator to determine whether the application of Na$^+$ and dopamine in the external buffer would influence K$^+$ flux. To follow the intra-vesicular level of K$^+$, the K$^+$ indicator was encapsulated in PLs containing 100 mM K$^+$ and 100 mM NMDG$^+$. Introduction of external buffer containing dopamine, 100 mM Na$^+$, and 100 mM K$^+$ trigged an exponential decrease in the signal from the encapsulated K$^+$ indicator, suggesting an efflux of K$^+$ ions (Fig. 5b). The persistent recording

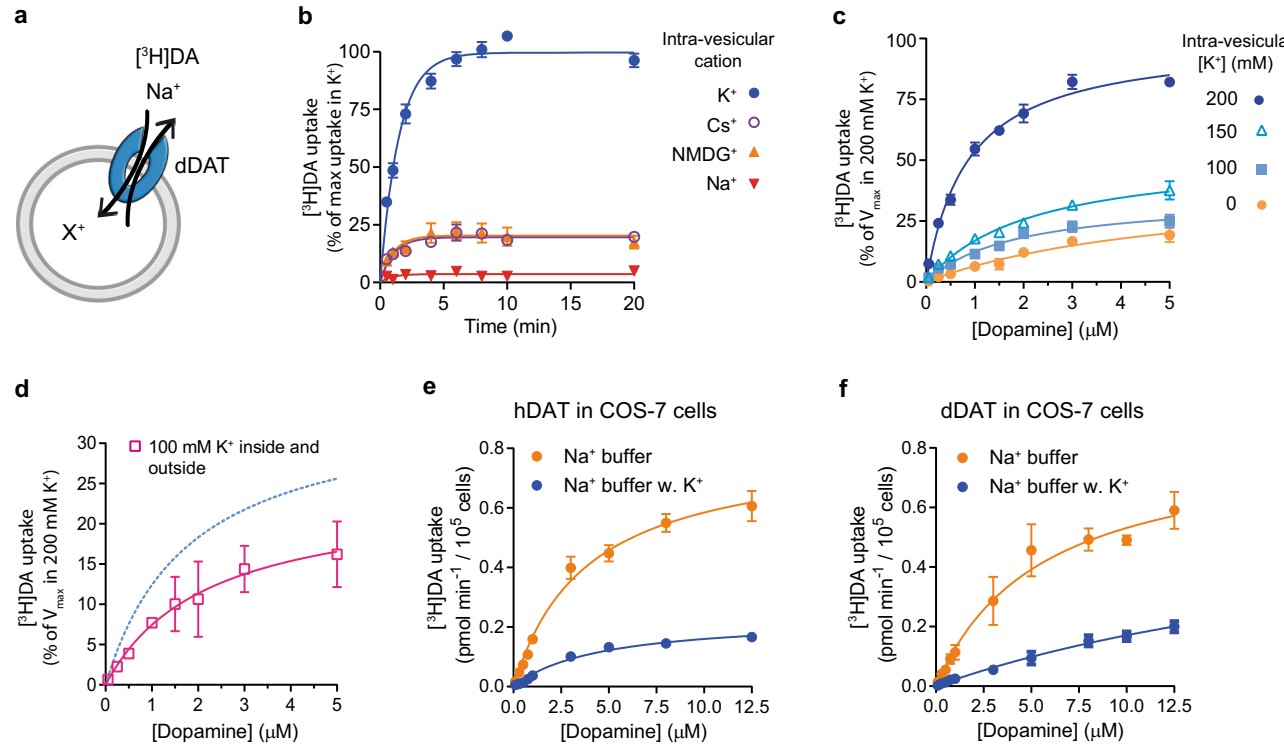

**Fig. 4 Dopamine transport is increased by intra-vesicular $K^+$ in proteoliposomes and decreased by $K^+$ in the uptake buffer. a** Schematic of the experimental setup for panel b, c and d. dDAT is reconstituted into PLs with the indicated monovalent cations in the intra-vesicular buffer ($X^+$). Transport is initiated by the addition of $Na^+$ and $[^3H]$dopamine to the extra-vesicular environment. **b** Time-dependent $[^3H]$dopamine uptake. Substitution of intra-vesicular $K^+$ with $NMDG^+$ or $Cs^+$ decreased the transport capacity ($n = 4$ (for $K^+$) or 3 independent experiments). Lines are fit to one-phase associations (Supplementary Table 4). **c** $[^3H]$dopamine saturation uptake into PLs with varying intra-vesicular $K^+$ concentration substituted with $NMDG^+$ to maintain the ionic strength at 200 mM. Lines are fit to Michaelis-Menten kinetics (Supplementary Table 5). Decreasing the intra-vesicular $K^+$ concentration significantly decreased $V_{max}$ determined from a Tukey's multiple comparisons test, the adjusted $p$-value for the comparison of 200 mM $K^+$ with 150 mM $K^+$ is $P = 0.006$, and for 200 mM $K^+$ compared to 100 mM $K^+$ $P < 0.0001$. $n = 4$ (for 200 and 0 mM $K^+$) or 3 independent experiments. **d** Same as (**c**) but with equal concentrations of $K^+$ (100 mM) inside the vesicles and in the uptake buffer ($n = 3$ independent experiments). Dissipation of the $K^+$ gradient significantly decreased the $[^3H]$dopamine uptake relative to the condition with an outward directed $K^+$ gradient (100 mM intra-vesicular $K^+$ from (**c**), indicated by the dashed blue line in the graph). This was determined by a two-sided unpaired $t$-test performed on the error-propagated $V_{max}$ estimates, $P = 0.0164$. **e** $[^3H]$Dopamine transport into COS-7 cells transfected with hDAT in a buffer containing 30 mM NaCl and either 105 mM NMDG-Cl (orange circles) or 105 mM KCl (blue circles). Lines are fits to Michaelis-Menten kinetics. In the presence of NMDG-Cl the $V_{max}$ and $K_m$ was $0.81 \pm 0.05$ pmol min$^{-1}$/10$^5$ cells and $4.53 \pm 0.86$ µM (mean ± SEM), respectively. In the presence of KCl, the $V_{max}$ and $K_m$ was $0.23 \pm 0.02$ pmol min$^{-1}$/10$^5$ cells and $3.84 \pm 0.57$ µM (mean ± SEM), respectively. By a two-sided unpaired $t$-test it was determined that there is no significant difference in the $K_m$ values, $P = 0.74$, and a significant decrease in $V_{max}$ with $K^+$ in the buffer, $P = 0.0034$. $n = 3$ independent experiments. **f** $[^3H]$Dopamine uptake in COS-7 cells transfected with dDAT performed in the same uptake buffers as in **e**. Lines are fits to Michaelis-Menten kinetics ($n = 4$ independent experiments). All independent experiments were performed in triplicates. Data points in (**b, c, d, e, f**) represent mean ± SEM (error bars). Data are provided for all experiments as a Source Data file.

of $K^+$ flux in the absence of a $K^+$ gradient supports the suggestion of a $K^+$ antiport. Surprisingly, the fraction of responding liposomes was in this case not affected by dopamine. However, the response was blocked by nortriptyline, suggesting that the $K^+$ efflux is dDAT-mediated (Fig. 5d).

**The rates of $Na^+$ and $K^+$ flux across the membrane are increased by dopamine.** The time-resolved nature of the fluorescence change in the single-vesicle setup allowed us to investigate the $Na^+$ and $K^+$ flux rates in individual active PLs. We first evaluated how $Na^+$ flux rates were affected by dopamine and $K^+$ in vesicles loaded with the $Na^+$ indicator. In the presence of intra-vesicular $NMDG^+$, dopamine did not significantly affect the $Na^+$ influx rate (Fig. 5e). In contrast, in vesicles containing $K^+$, the addition of dopamine significantly increased the $Na^+$ influx rate (Fig. 5f). When comparing the dopamine-dependent $Na^+$ influx rates between vesicles loaded with $NMDG^+$ and with $K^+$, we found that the $Na^+$ influx rates were increased markedly by $K^+$

(Fig. 5g). This suggests that the $Na^+$ influx associated with dopamine uptake is increased by intra-vesicular $K^+$. In the absence of dopamine, the rates of $Na^+$ flux were not different between vesicles containing either $K^+$ or $NMDG^+$ (Supplementary Fig. 7), suggesting that dopamine-independent $Na^+$ flux is unaffected by the identity of the intra-vesicular ion.

Next, we looked at the individual fluorescence traces from liposomes loaded with fluorescent $K^+$ indicator. In line with our observations from the $Na^+$ indicator, we here observed a significant increase in the rate of fluorescence intensity change from the $K^+$ indicator, suggesting an increase in $K^+$ efflux rates in the presence of dopamine (Fig. 5h). Taken together, the results from the single vesicle setup suggest that the number of vesicles responding with a $Na^+$ influx is increased with dopamine. For $K^+$ it is rather the rate of efflux and not the number of responding vesicles that are affected by dopamine. The correlation between the increase in rates of both $Na^+$ influx and $K^+$ efflux in the presence of dopamine is consistent with

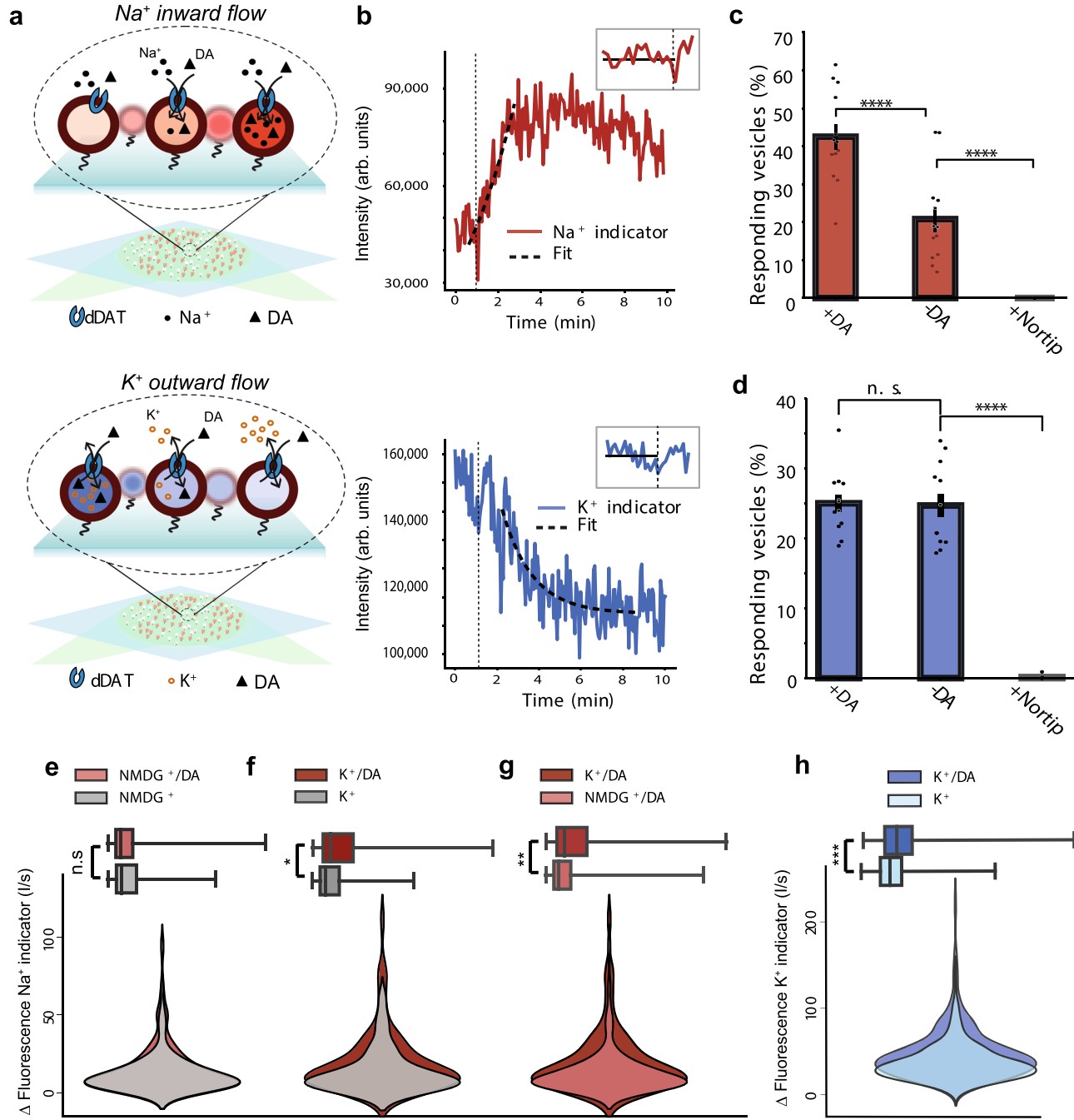

our suggestion that the coupling of $K^+$ to dopamine transport increases the rate of uptake.

## Discussion

Here, we have studied the effect of $K^+$ on the molecular, pharmacological, and functional properties of DAT and have shown real-time ion transport by dDAT, visualized in single vesicles resolution. Assays were performed with His-tagged full-length dDAT avoiding possible artifacts from deletions or substitution mutations or chemical protein modifications often necessary in biophysical assays. Across methods, our results are consistent with the suggestion that $K^+$ takes part in dDAT-mediated dopamine transport. Our Schild plot shows that $K^+$ binds to dDAT with an affinity of about 100 mM, which is within a physiological concentration range for resting neurons.

Our HDX-MS data revealed that $K^+$ induces a conformational state that is distinct from both the dDAT apo- and $Na^+$-state. $K^+$ decreases stability mainly on the intracellular face of dDAT and stabilizes several extracellular regions. The $K^+$ state is consistent with a mechanism in which $K^+$ binding promotes an inward-facing dDAT conformation that allows the release of $Na^+$ and substrate on the intracellular side. It is also in agreement with the proposed $K^+$ states of both SERT[34] and LeuT[20] as observed using HDX-MS. Functionally, intra-vesicular $K^+$ causes an increase in the uptake capacity and uptake rate of dopamine relative to vesicles containing intra-vesicular $Cs^+$ or $NMDG^+$, and this increase is unrelated to the level of active transporters in the vesicles. This suggests that the function of $K^+$ is to increase the rate of the rate-limiting step in the dDAT transport cycle or to inhibit the rebinding of $Na^+$ and dopamine, which otherwise can obstruct the progression of the transport cycle or even promote

**Fig. 5 Time-resolved Na$^+$ and K$^+$ flux from single dDAT proteoliposomes. a** Representation of single liposome measurements with TIRFm of PLs containing Na$^+$ indicator (top) or K$^+$ indicator (bottom). **b** Representative fluorescence intensity trace from the Na$^+$ indicator (top) and K$^+$-indicator (bottom) from a single vesicle after addition (inset and dashed line) of uptake buffer. **c** The percentage of vesicles showing fluorescence increase in the Na$^+$ indicator signal after addition of uptake buffer without dopamine (-DA) 20.8 ± 3.3% and with dopamine (+DA) 42.5 ± 3.3% (mean ± SEM). Addition of nortriptyline (+Nortrip) to the uptake buffer inhibited the response. The bar plots are represented as mean ± SEM. Each data point represent the percentage of responding PLs from a chamber surface containing approximately 300 PLs ($n = 12$ independent experiments). **d** As in (**c**) but shown for vesicles containing the K$^+$ indicator responding with a decay in fluorescence. Responding vesicles was 25.0 ± 4.3% with dopamine and 24.7 ± 5.7 % without. No responses were observed when nortriptyline (+Nortrip) was added. The bar plot represents mean ± SEM ($n = 12$ independent experiments). Significance was determined by a two-sided KS test and a two-sided $t$-test: not significant (n.s.) indicates $P > 0.05$, ****$P < 0.0001$. **e–g** Bottom panels: Overlays of violin plots displaying the probability distribution of the rates of fluorescence intensity change (I/s) from the Na$^+$ indicator from single PLs containing either NMDG$^+$ or K$^+$, measured with and without DA in the uptake buffer. The width of the plots display the kernel density of the rates. The number of single liposomes assessed in each condition is summarized in Supplementary Table 7. The liposomes were assayed in 12 independent experiments for each condition. Top panels: Box plots showing the statistical dispersion of the fluorescence increase rates from the violin plots. The box plots are defined with median (med), quartile (25%) Q1 (q1), quartile (75%) Q3 (q3) and upper whiskers equal to upper bound and maxima and lower whiskers equal to lower bound and minima to show the full distribution. n.s. not significant; *, $P < 0.05$; **, $P < 0.01$. The statistical significance of the difference between the distributions in the violin plots was evaluated with a two-sided KS-test. For the violin plots in (**e**) n.s., $P = 0.74$, in (**f**) $P = 0.044$, in (**g**) $P = 0.0069$. **h** Overlay of violin plots and bar plots as in (**e–g**) of the rates of fluorescence signal decrease obtained from the K$^+$ indicator with and without DA in the uptake buffer. ***, $P < 0.001$. Two-sided KS-test for violin plots, $P = 1.25*10^{-8}$. The values for max, min, med, q1 and q3 from the boxplots in (**e–h**) are summarized in Supplementary Table 7. Data are provided as a Source Data file.

reverse transport. The latter has been suggested as the role for K$^+$ in LeuT[19]. It is also possible that the ability of the K$^+$ gradient to drive accumulation of dopamine is caused by uncoupled efflux of K$^+$ through dDAT, which become stimulated during Na$^+$ coupled uptake by the accumulation of positive charge inside the vesicles.

The properties of dDAT are expected to be highly sensitive to the lipid composition of the detergent-lipid micelles and the liposomes not least because of the cholesterol binding sites observed in the transporter[24,35], thus it is possible that the magnitude of the effect of the ions reported here would be different in a different lipid composition. Nevertheless, an effect of K$^+$ on uptake of dopamine does appear also to be present in COS-7 cells transiently transfected with both dDAT and hDAT, which represents a more native membrane environment for the transporters than liposomes.

We cannot rule out that a fraction of the liposome-reconstituted dDAT is in the inside-out orientation. However, when imposing an inward-directed Na$^+$ gradient we promote unidirectional transport. We have assessed the dopamine and ion transport in PLs in bulk and in single vesicle resolution but without knowing the exact number of active transporters or their distribution in the vesicles. To desiccate further the uptake kinetics for dDAT it would be useful to label the transporter to be able to assess transport in single molecule resolution.

The reconstitution in PLs also allows the encapsulation of fluorescent sensors such as the Na$^+$ and K$^+$ indicators. We were able to visualize the movement of ions across the membrane by dDAT in the single vesicles. We saw that Na$^+$ and K$^+$ transport could be uncoupled from dopamine transport. Both hDAT and SERT have previously been shown to possess both uncoupled currents and leak currents not associated with transport[41,42]. The constitutive dopamine-independent flux of Na$^+$ and K$^+$ we observe could be parallel to the leak currents measured by electrophysiology[42]. When dopamine was added externally, it increased the rate by which K$^+$ was antiported. When both dopamine and K$^+$ was present on the external and intra-vesicular side respectively, the rate of Na$^+$ fluorescence change increased, and correlated with a faster decrease in K$^+$ fluorescence. The observation is consistent with a transport mechanism where Na$^+$ and dopamine uptake became coupled with K$^+$ antiport.

Our results suggest that K$^+$ antiport is a significant component of the dopamine reuptake mechanism of dDAT. It is permissive but not obligate for dopamine transport. The effects observed

here align with observations on SERT[16] and LeuT[19,20], linking our results to both an evolutionary close - and a distantly related NSS, respectively. Other papers have suggested that K$^+$ influences compound binding to hDAT[43–45]. The pharmacological data and the transport data from COS-7 cells we present here on hDAT opens the possibility that K$^+$ could have a similar effect in both *Drosophila* and human dopamine transport. Taken together, this suggests that K$^+$ antiport is more common among the NSS family of transporters than previously anticipated and could be a rule rather than the exemption. In the broader perspective, understanding the NSS transport mechanism in molecular detail forms part of the basis for guiding the development of drugs that modulate the transporters. Furthermore, it could be important for precision medicine by allowing prediction of the functional implications of transporter single nucleotide polymorphism in individuals suffering from neuropsychiatric disorders[4,46] and thus guide the choice of drugs for treatment.

## Methods

**Constructs.** Full-length dDAT with a C-terminal thrombin cleavage site followed by an 8-His-tag synthesized by GenScript Inc. (Piscataway, NJ)[23] cloned into the pEG BacMam expression vector[28]. Full-length hDAT with an N-terminal 12-His-tag followed by a thrombin cleavage site synthesized by GenScript Inc. (Piscataway, NJ) cloned into the pEG BacMam expression vector. Full-length hDAT cloned into the pcDNA3.1 vector (ThermoFisher Scientific) for expression in COS-7 cells.

**Protein expression and purification.** Full-length dDAT-8-His was expressed by transduction of Expi293F suspension cells (ThermoFisher Scientific) with baculovirus produced in Sf9 cells (Expression Systems). Cells were harvested by centrifugation (6000 x g) and membranes were prepared by homogenization and sonication (Branson Sonifier 250, 50% duty cycle, power setting 2). Membranes were pelleted at 125,000 x g for 3 h. The pelleted membranes were homogenized and solubilized in buffer containing 20 mM Tris, pH 8.0, 150 mM NaCl, 20 mM *n*-dodecyl β-D-maltoside (DDM), 4 mM cholesteryl hemisuccinate (CHS), 5 μg/ml benzamidine, and 10 μg/ml leupeptin. The detergent-solubilized dDAT was incubated with His-Pur Ni-NTA resin (SigmaAldrich) for 2 h. The resin was washed in buffer A containing 40 mM Tris, pH 8.0, 300 mM NaCl, 5% glycerol, 14 μM lipids (1-palmitoyl-2-oleoyl-sn-glycero-3-phosphocoline (POPC), 1-palmitoyl-2-oleoyl-sn-glycero-3-phosphoethanolamine (POPE), and 1-palmitoyl-2-oleoyl-sn-glycero-3-phospho-(1'-rac-glycerol) (POPG) at a weight ratio of 3:1:1), 1 mM DDM, 0.2 mM CHS supplemented with 5 μg/ml benzamidine and 10 μg/ml leupeptin in a gradient of imidazole on a column by gravity flow. The protein was eluted from the affinity column with buffer A supplemented with 300 mM imidazole. The protein was spin-concentrated and loaded on a FPLC size exclusion column (Superose$^{TM}$ 6 Increase 10/300 GL, Cytiva) to increase purity and remove imidazole by buffer exchange into buffer A. Finally, the protein was spin-concentrated and stored at −80 °C for further use. All purification procedures were performed on ice or at 4 °C.

Full-length hDAT was expressed using the same approach as for dDAT. The cells were harvested by centrifugation (6000 x g), and membranes were prepared by sonication (Branson Sonifier 250, 50% duty cycle, power setting 3) in buffer (30 mM NaHEPES, pH 8.0, 30 mM NaCl, 5 mM KCl, 5 mM Ethylenediaminetetraacetic acid (EDTA), 7 mM MgCl₂, 10% (w/v) sucrose, 5 μg/ml benzamidine, 10 μg/ml leupeptin, 1:1000 (v/v) Protease Inhibitor Cocktail (Sigma), 2 μg/ml DNase I, and 2 μg/ml RNase A). The membranes were pelleted by centrifugation (100,000 x g) and washed in buffer containing 30 mM NaHEPES, pH 8.0, 1 M NaCl, 5 μg/ml benzamidine, 10 μg/ml leupeptin, Protease Inhibitor Cocktail (Sigma), and 10 mM DTT by homogenization. This step was repeated twice. The membranes were homogenized in buffer containing 30 mM NaHEPES, pH 8.0, 30 mM NaCl, 5 mM KCl, 10% (w/v) sucrose and 10 mM DTT, and stored at −80 °C. All steps were performed on ice or at 4 °C.

**[³H]Nisoxetine binding.** Na⁺-dependence of [³H]nisoxetine binding to dDAT was determined using 0.5 μg ml⁻¹ of purified dDAT mixed with 12 nM [³H] nisoxetine (80.4 Ci mmol⁻¹, PerkinElmer), 108 nM nisoxetine, and buffer containing 20 mM Tris-Hcl, pH 8.0, 0.05% (w/v) DDM, 0.01% (w/v) CHS, 14 μM lipids (3:1:1, POPC:POPE:POPG)) supplemented with the indicated NaCl and KCl concentrations. The ionic strength was maintained by substituting Na⁺ and K⁺ with NMDG⁺. The indicated conditions were added to a clear-bottom, 96-well plate (Corning) and incubated with agitation at 4 °C. 5% (v/v) YSi-Cu His-Tag SPA beads (Perkin Elmer) was added to each well. Plates were sealed, mixed at RT on an agitator and left to settle at RT for 2 h. Plates were counted on a 2450 MicroBeta² microplate counter (PerkinElmer). Each independent experiment was performed in triplicates. The experiments were performed with dDAT from two independent purifications.

**[³H]MFZ 2-12 binding.** Na⁺-dependence of [³H]MFZ 2-12 binding to hDAT was determined using 400 μg (wet-weight) of Expi293F membranes expressing hDAT mixed with 30 nM [³H]MFZ 2-12 (30 Ci/mmol, American Radiolabeled Chemicals, Inc.) in a buffer containing 10 mM NH₄PO₄, pH 8.0, supplied with the indicated NaCl and KCl concentrations. The ionic strength was maintained by substituting Na⁺ and K⁺ with NMDG⁺. The indicated conditions were added to a clear, round-bottom, 96-well plate (Corning) and mixed briefly at RT on an agitator. The plate was sonicated for 30 s in a bath sonicator and incubated for 90 min at RT on an agitator. To determine [³H]MFZ 2-12 binding to hDAT, the 96-well plate was placed on a cell harvester (Tomtec harvester 96 match II) and the membranes were trapped on a 96-well glass fiber filter (Filtermat B – GF/B, Perkin Elmer) soaked in 1.5% poly(ethyleneimine) solution. The glass fiber filter was washed with 1 liter ice-cold buffer containing 10 mM NH₄PO₄, pH 8.0, and 1 M NaCl. Hereafter, the filter was dried and MeltiLex B/HS (Perkin Elmer) was melted onto the filter. The filter was counted in a 2450 MicroBeta2 microplate counter (PerkinElmer). Non-specific binding was determined by addition of 2 μM nomifensine. Each independent experiment was performed in triplicates. The experiments were performed with membranes from two independent preparations.

**Data analysis of radiolabelled substrate binding.** Data describing the Na⁺-dependence of radioligand binding to dDAT and hDAT in varying K⁺ concentrations were fitted to a sigmoidal dose-response (variable slope) equation, otherwise known as a four-parameter logistic equation (Prism 9.0, GraphPad Software). This fit derived potency estimates (EC₅₀ values) i.e., the [Na⁺] concentration that displays half-maximal [³H]nisoxetine or [³H]MFZ 2-12 binding, according to the upper (Bₘₐₓ) and lower plateaus of the sigmoid. Data were normalized to the top and bottom plateau for each K⁺ condition. For dDAT data, Schild plots were calculated by taking the ratio between the EC₅₀ values for Na⁺ in the presence and absence of the antagonist, here K⁺, producing the dose ratios (dr) at the different K⁺ concentrations ([B]) used in Eq. (1)[30]. Values of log(dr-1) were plotted as a function of their corresponding log[B] values. The affinity of K⁺ for dDAT (K_B) was determined using the Schild Eq. (1):

$$\log(dr - 1) = \log[B] - K_B \qquad (1)$$

**Hydrogen-deuterium exchange mass spectrometry.** The K⁺ state was induced by dialyzing affinity-purified dDAT against buffer containing 40 mM Tris, pH 8.0, 200 mM KCl, 5% glycerol, 14 μM lipids (POPC, POPE, and POPG at a weight ratio of 3:1:1), 1 mM DDM, and 0.2 mM CHS at 4 °C. Dialyzed dDAT was equilibrated at a concentration of 1.5 μM at 25 °C for 30 min. Deuterium exchange was initiated by diluting the protein samples 1:4 with 94% D₂O buffer 40 mM Tris, pH 8.0, 200 mM KCl, 5% glycerol, 14 μM lipids (POPC, POPE, and POPG at a weight ratio of 3:1:1), 1 mM DDM, 0.2 mM CHS at 25 °C. At defined time points (0.25 min, 1 min, 10 min, 60 min, and 480 min), the labeling reaction was quenched by mixing (1:1) aliquots of 15 pmol dDAT with ice-cold quench buffer (220 mM phosphate buffer, pH 2.3, 2 M urea). Samples were stored at −80 °C until further use. All time points were performed as recommended[47] in three technical replicate measurements (n = 3). The isotopic exchange was performed simultaneously with the Cs⁺ and Na⁺ states as well as the maximum-labeled control samples reported previously[23]. The maximum-labeled control samples were prepared from pre-digested dDAT using the same conditions as for the HDX experiments. Following

online pepsin digestion, dDAT peptides were desalted, eluted, and collected manually in a single fraction and lyophilized. 75.2% deuterated buffer containing 40 mM Tris, pH 8.0, 200 mM NaCl, 5% glycerol, 14 μM lipids (POPC, POPE, and POPG at a weight ratio of 3:1:1), 1 mM DDM, and 0.2 mM CHS was used to reconstitute the dDAT peptides. To reach full deuteration, peptides were incubated in the deuterated buffer for 24 h at 25 °C before quenching.

Prior to mass analysis, quenched samples (15 pmol) were rapidly thawed and injected into a cooled (0 °C) nanoACQUITY UPLC HDX system (Waters). Protein samples were digested online at 20 °C on an in-house packed immobilized pepsin column prepared using pepsin agarose resin (Thermo Fisher scientific). The resulting peptides were desalted on a C8 trap column (VanGuard pre-column ACQUITY UPLC BEH C8 1.7 μm, Waters) for 3 min at a flow rate at 200 μl/min solvent A (0.23% formic acid in water, pH 2.5) and separated by reversed-phase chromatography over a C8 analytical column (ACQUITY UPLC BEH C8 1.7 μm, 100 nm, Waters) with a C8 trap column in front using a linear gradient from 8–30% solvent B (0.23% formic acid in acetonitrile) over 10 min at a flow rate of 40 μl/min. Mass analysis was conducted on a hybrid Q-TOF SYNAPT G2-Si mass spectrometer (Waters) equipped with a standard ESI source operated in positive ion mode. Mass spectra were lock-mass corrected against Glu-fibrinopeptide B. Ion mobility separation was used to enhance peak capacity and minimize spectral overlap. The ion mobility cell was operated using a constant nitrogen flow of 90 ml min⁻¹ at a wave velocity of 580 m s⁻¹ and a wave height of 40 V.

HDX-MS data for the K⁺ state were acquired simultaneously with the Cs⁺ and Na⁺ states previously reported[23]. Maximum-labeled control samples shown here are from our previous study[23], and were processed and analyzed using the HDX-MS workflow described above except for the pepsin column, which was removed to avoid additional digestion. Peptide identification from non-deuterated samples were performed by tandem mass spectrometry (CID) using a combination of data-independent acquisition (DIA) and data-dependent acquisition (DDA) with the ion mobility cell turned off.

**HDX-MS data evaluation and statistical analysis.** The deuterium exchange levels were calculated for all identified peptides using DynamX 3.0 (Waters) with manual verification of all peptide assignments. Noisy and overlapping spectral data were discarded from the HDX-MS analysis. To allow for quantitative comparison of samples not measured on the same day, back-exchange was calculated from the maximum-labeled control samples for all individual peptides according to Eq. (2).

$$BE(\%) = \left(1 - \frac{m_{75.2\%} - m_{0\%}}{m_{MAX} - m_{0\%}}\right) \cdot 100\% \qquad (2)$$

where $m_{75.2\%}$ denotes the mass of the maximum-labeled peptide, $m_{0\%}$ is the mass of the non-deuterated peptide, and $m_{MAX}$ denotes the theoretical maximum deuterium uptake of the peptide (excluding N-terminus and prolines). BE(%) was used to normalize deuterium uptake values between measuring days to allow for a quantitative comparison of samples not measured on the same day. The deuterium uptake of individual states were compared for all identified peptides in Microsoft Excel (Microsoft) using either a homoscedastic or a heteroscedastic Student's t-test (α = 0.01) depending on an F-test (α = 0.05) that compared the variance of deuterium uptake from two different states for each single peptide at a single time point. For each peptide, a difference in HDX between two states was only considered significant if two consecutive time points showed a significant difference in deuterium uptake (p < 0.01). Furthermore, differences in HDX between the Cs⁺ and K⁺ states and the Na⁺ and K⁺ states had to exceed threshold values of 0.24 and 0.20 deuterons (D), respectively, which corresponded to the 95% confidence interval (CI) calculated according to Eq. (3).

$$CI = \bar{x} \pm t \cdot \frac{\sigma}{\sqrt{n}} \qquad (3)$$

here $\bar{x}$ is the average difference in deuterium content assuming a zero-centered distribution (x = 0), t is 4.303 for the 95% CI with 2 degrees of freedom, σ is the pooled propagated standard deviation of differences in deuterium content for all peptides across all time points (i.e., 0.25–480 min) for the two states compared, and n is the number of replicate samples (n = 3). The HDX data presented were qualified against an exploratory dataset obtained from a dDAT sample expressed and purified from a separate batch of Expi293F cells. This HDX-MS experiment was preliminary in nature and served to explore the HDX of dDAT with or without Cs⁺ and K⁺ and Na⁺. The second HDX-MS experiment was slightly optimized based on the exploratory experiment and was more comprehensive and included more technical replicate measurements. Overall, there was a good qualitative agreement between the two HDX-MS experiments and similar trends were observed pertaining to the impact of Cs⁺ and K⁺ and Na⁺ on dDAT. Due to minor differences in the experimental details between the experiments, a direct quantitative comparison should not be performed. We have included the data from the exploratory HDX-MS experiment in Supplementary Fig. 8 for comparison. All results included in this paper as well as the statistical analysis of error is based on data from the second experiment. HDX results were mapped onto the dDAT crystal structure (PDB ID: 4XP1) using PyMOL (The PyMOL Molecular Graphics System, Version 2.0 Schrödinger, LLC).

The HDX Summary Table (Supplementary Table 2) and the HDX Data Table (Supplementary Table 3) are included according to the community-based recommendations for HDX data availability[47].

**Reconstitution of dDAT into liposomes.** Liposomes were prepared from aso-lectin:cholesterol:brain polar lipid extract stored in chloroform (Avanti) to reach a molar ratio of 60:17:20. The lipid mix was dried under a stream of $N_2$ for 2 h. The lipid film was suspended in buffer containing 20 mM HEPES, pH 7.5, and 200 mM KCl to 10 mg lipid/ml by alternating vortexing and bath sonication. The liposomes went through five freeze-thaw cycles, and were extruded with a mini extruder (Avanti) through a Nuclepore™ Track-Etch Membrane polycarbonate filter with 400 nm pore size (GE Healthcare Life Sciences). The liposomes were diluted to 4 mg lipid/ml and mixed with purified dDAT (concentrated to 2 mg/ml) in a 1:150 protein:lipid (wt/wt) ratio. After 30 min incubation under slow rotation at 7 °C, SM-2 bio-beads (Bio-Rad Laboratories) equilibrated in buffer containing 20 mM HEPES, pH 7.5, and 200 mM KCl were added in a 35 mg (semi-dry weight)/ml beads to buffer ratio. SM-2 bio-beads were added again after 30 min, after 60 min and after 15 h at 7 °C under slow rotation. After the last addition, beads and proteoliposomes (PLs) incubated together for 2 h. Bio-beads were removed by filtration. The PL sample was split into centrifugation tubes and diluted 25 times in the final intra-vesicular buffers containing 200 mM chloride salt (either NMDG-Cl, CsCl, KCl or NaCl) and 20 mM HEPES, pH 7.5, and centrifuged at 160,000 x g for 1 h at 4 °C. Subsequently, the pelleted PLs were re-suspended to 10 mg lipid/ml in their final intra-vesicular buffer, and frozen in liquid $N_2$ in appropriate aliquots until use. PLs for TIRFm were made as described with the following additions to the protocol. In the lipid mix 1:1000 molecular ratio of biotinylated PEG-DOPE (Biopharma PEG) and ATTO-655-DOPE membrane dye (ATTO-TEC) was added. At all times, the sample was shielded from light to protect the fluorophores. After two freeze-thaw cycles the liposome sample was split in two, and 10 mM Na$^+$-sensor (Sodium Green™, Tetra (Tetramethylammonium, ThermoFisher) Salt or 10 mM K$^+$-sensor (ION Potassium Green-2 TMA+ Salt, Abcom) was added.

**[³H]Dopamine uptake experiments with proteoliposomes.** PLs were thawed and extruded through Nuclepore™ Track-Etch Membrane polycarbonate filters with 400 nm pore size (GE Healthcare Life Sciences). The uptake assay was performed in 96-well ultra-low attachment, round bottom plates (Costar) at 25 °C. Nonspecific binding and nonspecific uptake across the lipid bilayer was measured using PLs incubated with 100 µM nortriptyline for 20 min prior to the uptake. To start the uptake experiment, the PLs were diluted to a final concentration of 0.5 mg lipid/ml in freshly made uptake buffer containing 20 mM HEPES, pH 7.5, 1 mM EDTA, 1 mM L-ascorbic acid supplemented with [³H]dopamine (PerkinElmer). The total reaction volume was 200 ul. For time-dependent uptake, the buffer also contained 200 mM NaCl, and for concentration-dependent uptake it also contained 100 mM NaCl and 100 mM NMDG-Cl. For time-dependent uptake the buffer was supplemented with 0.5 µM [³H]dopamine (4.54 Ci/mmol). For concentration-dependent uptake the buffer contained [³H]dopamine (5.14 Ci/mmol) from [50–500 nM] and 1.542 Ci/mmol from [1–5 µM]. For concentration-dependent uptake, the reaction time was 3 min, which allowed an approximate measure of the initial rate at each substrate concentration to be obtained. To terminate the reaction, the PLs were trapped on a 96 well glass fiber filter (Filtermat B – GF/B, Perkin Elmer) pre-soaked in 1.5% poly(ethyleneimine) using a Tomtec harvester (96 match II). Each filter position was washed with 1 ml freshly made, cold wash buffer (20 mM HEPES, pH 7.5, 200 mM KCl, 1 mM EDTA, 1 mM L-ascorbic acid). Hereafter, the filter was dried and MeltiLex B/HS (Perkin Elmer) was melted onto the filter. The filter was counted in a 2450 MicroBeta² microplate counter (PerkinElmer). Each independent experiment was conducted in triplicates. All experiments were performed with dDAT from at least two independent purifications and PLs from at least two independent reconstitutions unless otherwise stated.

**Measuring relative amount of active transporters in proteoliposomes.** From the extruded liposomes with different intra-vesicular buffers used in the uptake experiments describes above, a sample was taken from each condition and diluted 25x in buffer A supplemented with 15% glycerol, 1% DDM, 4 mM CHS for 15 h to dissolve the PLs and re-suspend the protein in detergent at 7 °C. The detergent suspended dDAT was diluted 10x in buffer A supplemented with 600 nM [³H] nisoxetine (4.02 Ci/mmol, PerkinElmer) and 7% (v/v) YSi-Cu His-Tag SPA beads (Perkin Elmer) binding to dDAT via the His-tag. [³H]Nisoxetine binding was measured on a 2450 MicroBeta² microplate counter after 30 min incubation at room temperature and again after 24 h incubation at 7 °C. Total binding was determined in triplicates. Non-specific binding was measured in the presences of a saturating nortriptyline concentration (200 µM) and subtracted from the counts per min (c.p.m.) measured in total binding. The specific maximum binding was regarded as a relative measure of the number of active transporters in each condition, and was used to normalize the uptake data to the amount of active transporters in the given condition within each uptake experiment. Based on the assumption that the counting efficiency of the 2450 MicroBeta² microplate counter was 50% in SPA counting mode the $B_{max}$ values obtained could be converted to an estimate of the amount of active dDAT in the sample (Supplementary Table 6).

**Data analysis [³H]dopamine uptake.** For time-dependent dopamine uptake, the data were normalized to dDAT activity from each condition. Non-specific [³H] dopamine binding measured in the presence of nortriptyline was subtracted. Normalized specific uptake data were fitted to a one phase association as Eq. (4) using GraphPad Prism 7.0.

$$y(t) = p * e^{(-k*t)} \qquad (4)$$

where $p$ is the maximal uptake (the amplitude) and $k$ is the rate constant in min$^{-1}$, $t$ is time in min and $y$ is uptake in c.p.m. after normalization to transporter concentration. Subsequently, data from each experiment were normalized to the value of $p$ from the condition with the highest maximal uptake, which allowed us to compare uptake experiments from different reconstitutions. The normalized data sets from each experiment were combined and re-fitted to the one phase association.

For concentration dependent uptake, the data were normalized to dDAT activity from each condition. Non-specific [³H]dopamine binding measured in the absence of a Na$^+$ gradient was subtracted. Normalized specific uptake data were fitted to the Michaelis-Menten equation. Subsequently, the data were normalized to the $V_{max}$ value determined from the condition with the highest $V_{max}$. The normalized data sets from each experiment were combined and re-fitted to the Michaelis-Menten equation.

**[³H]Dopamine transport in intact COS-7 cells.** COS-7 cells were transiently transfected using lipofectamine with 0.35 µg DNA of hDAT in the pcDNA3.1 vector or 1 µg DNA of dDAT in pEG BacMam vector per 10⁶ cells in a lipo-fectamine:DNA ratio of 3.5. Cells were seeded in 24-well plates (5*10⁴ per well) coated with poly-ornithine and incubated in 10% CO₂ at 37 °C. [³H]dopamine transport assays were performed 48 h after transfection. The transport experiments were carried out at room temperature. Prior to the experiment, each well were washed with 450 µl uptake buffer at pH 7.4 containing 30 mM NaCl, 25 mM HEPES, 1.2 mM CaCl₂, 1.2 mM MgSO₄, 1 mM ascorbic acid, and 5 mM glucose supplemented with either 105 mM NMDG-Cl or 105 mM KCl. Uptake was performed in the indicated uptake buffer by addition of [³H]dopamine (0.6 Ci/mmol, PerkinElmer) [0.1; 12.5 µM] in a total volume of 500 µl. After 7.5 min the uptake reaction was stopped by two washes with 500 µl ice-cold uptake buffer per well. Cells were lysed in 1% sodium dodecyl sulfate (SDS) for 60 min at 37 °C, then transferred to 24-well sample plates (PerkinElmer) and counted in a 2450 MicroBeta² microplate counter (PerkinElmer, Waltham, MA) after addition of 500 µl Opti-phase Hi Safe 3 scintillation fluid (PerkinElmer). All experiments were performed in triplicates. Non-specific uptake was determined in cells pre-incubated with either 10 µM nomifensine (hDAT) or 100 µM nortriptyline (dDAT). Specific uptake data were fitted to the Michaelis-Menten equation in GraphPad Prism 7.0.

**TIRF Microscopy setup.** All single PL experiments were acquired using an inverted total internal reflection fluorescence microscope model IX83 (Olympus). The microscope setup was equipped with an EMCCD camera model imagEM X2 (Hamamatsu) and a 100x oil immersion objective model UAPON 100XOTIRF (Olympus) and an emission quad band filter cube, as to block out laser light in the emission pathway. All data were recorded using three solid state laser lines (Olympus) at 488 nm, 532 nm and 640 nm in order to excite ATTO-655-DOPE membrane dye and the two ion indicators Potassium Green-2 and Sodium Green™. The data were acquired using a penetration depth at 200 nm (corresponding to angles of 59.97, 60.23 and 60.78 degrees for 488 nm, 532 nm and 640 nm laser lines respectively), with a dynamic range of 16-bit grayscale and an image dimension of 512 times 512 pixels, corresponding to a physical field of view length of 81.92 µm. A peristaltic flow-pump was installed on top of the TIRF setup, as to automatically flow in both dopamine, inhibitor and uptake buffers during the real time recordings.

**TIRF recordings of single proteoliposome assay.** Glass surfaces were prepared using plasma cleaned glass slice with fastened sticky-Slide VI 0.4 (Ibidi) and functionalized using PLL-g-PEG and PLL-g-PEG-biotin in a 100 to 1 ratio followed by a neutravidin layer[36,37]. Each glass surface contains 6 chambers that are independently utilized for liposomes immobilization and imaging. The liposomes were flowed into the selected microscope chamber using the pump setup for immobilization to obtain a liposome density of 200–250 liposomes per field of view. Unbound liposomes were washed away with 3x chamber volumes of buffer. Using CellSens imaging software (Olympus), the image recordings were fully automated. Briefly, we recorded a 6-field of view in parallel with a temporal resolution of 3.7 s/ cycle for Na$^+$ uptake and 3.4 s/cycle for K$^+$ outflow with a total of 700 cycles per experiment corresponding to a total experimental timeframe of 43–40 min, respectively. Real-time recording was started with the same buffer in the flow cell as inside the PLs, then the buffer was exchanged using the pump for the 0.5 mL of the respective uptake buffer with a flowrate of 0.5 mL/min. All experiments were performed with dDAT from at least two independent purifications and PLs from at least two independent reconstitutions.

**Compositions for uptake and intra-vesicular buffers and substrate concentration for TIRF recording.** Monitoring the outward K$^+$ transport, the intra-

vesicular buffer was composed of 10 mM ION Potassium Green-2 TMA+ Salt, 20 mM HEPES, pH 7.5, 100 mM KCl and 100 mM NMDG-Cl. The uptake buffer contained 20 mM HEPES, pH 7.5, 100 mM KCl, 100 mM NaCl, 1 mM L-ascorbic acid and 1 mM EDTA. Monitoring the inward transport of Na$^+$, two populations of liposomes with two distinct intra-vesicular buffers were prepared as to observe the dependence of K$^+$. Intra-vesicular buffer with K$^+$ was composed of: 10 mM Sodium Green™, Tetra (Tetramethylammonium) Salt, 20 mM HEPES, pH 7.5, 100 mM KCl and 100 mM NMDG-Cl. The intra-vesicular buffer without K$^+$ contained 10 mM Sodium Green™, Tetra (Tetramethylammonium) Salt, 20 mM HEPES, pH 7.5, 200 mM NMDG-Cl. The uptake buffer was composed of 20 mM HEPES, pH 7.5, 200 mM NaCl, 1 mM L-ascorbic acid and 1 mM EDTA. The concentration of dopamine in the uptake buffers was 0.5 μM, while the concentration of the inhibitor nortriptyline was 500 μM. All buffers were freshly prepared.

**Tracking and co-localization software for TIRF multiplexing experiments**. All individual liposomes were localized and tracked using the membrane signal from the ATTO-655-DOPE membrane dye and co-localized to the ion indicator signal. Tracking and localization was done using in-house developed python software based on previous publications[37,38] to ensure a nanometer precise localization and co-localization whilst simultaneously correcting for any potential x and y drift introduced by the pump. Using an adapted version of previously published software[37], the background corrected signal from both the membrane dye and ion indicator were extracted to ensure correct signal integration and adjusting potential bias from the known uneven TIRFm illumination profile (see Supplementary Fig. 9a, b). To provide local background corrected signal, a circular area (97 pixels) is used to integrate the signal from each individual liposome, where an outer annulus (88 pixels) form the basis for background evaluation.

**Fitting single data to single exponential decay**. After signal extraction, all active traces were found by inspection and the corresponding translocation rates determined using a maximum likelihood fitting scheme[36,38]. For K$^+$ experiments, the decreasing intensity traces were fitted with Eq. (5).

$$I(t) = p * e^{(-k*t)} + y \qquad (5)$$

where $p$ is the amplitude, $k$ is the rate constant and $y$ represents the offset from zero. For Na$^+$ experiments, the increasing intensity traces were fitted with Eq. (6).

$$I(t) = p * e^{(k*t)} + y \qquad (6)$$

where $p$ is the amplitude, $k$ is the rate constant and $y$ represents the offset from zero.

**Reporting summary**. Further information on research design is available in the Nature Research Reporting Summary linked to this article.

## Data availability

The data that support this study are available from the corresponding author upon reasonable request. Pharmacology data, bulk proteoliposome data, and cell uptake data generated in this study are provided in the Source Data file. HDX-MS data generated in this study are available via ProteomeXchange with identifier PXD032663. The single liposomes data generated in this study have been deposited in the UCPH erda database and are accessible here at https://sid.erda.dk/sharelink/BWguZqT2kL. Source data are provided with this paper.

## Code availability

Software code made for treatment of single vesicle data is available here: https://github.com/hatzakislab/Dopamine-Manuscript.

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

## Acknowledgements

We would like to thank Lone Rosenquist for technical assistance. We also thank for the review from Biophysics Colab. The work was supported in part by The Independent Research Fund Denmark (7016-00272A and 1030-00036B to C.J.L.), The Lundbeck Foundation (R344-2020-1020 to C.J.L.), The Novo Nordic Foundation (NNF19OC0058496 to C.J.L.), The Carlsberg Foundation (CF20-0345 to C.J.L. and CF16-0797 to N.S.H.) and The Villum Foundation (BioNEC grant 18333 to M.G.M. and N.S.H.). Work at The Novo Nordisk Foundation Center for Protein Research is funded in part by The Novo Nordisk Foundation (NNF14CC0001).

## Author contributions

C.J.L. conceptualized the idea. S.G.S., M.G.M., and A.K.N. designed the experiments together with K.D.R., N.S.H. and C.J.L. S.G.S., M.G.M., A.K.N., S.S-R.B., C.F.P., J.C.N., and I.H.P. performed the experiments and data analysis. All authors were involved in data interpretation. S.G.S. and C.J.L. prepared the manuscript with significant contribution from all authors.

## Competing interests

The authors declare no competing interests.
