## [Peer Review File · Nature Communications]

The dopamine transporter antiports potassium to increase the uptake of dopamineREVIEWER COMMENTS

Reviewer #1 (Remarks to the Author):

In this article, Schmidt and colleagues investigate the role of K⁺ ion on the transport mechanism of the NSS neurotransmitter dDAT. They first show that K⁺ competes with Na⁺ for binding to dDAT, then use HDX-MS to illustrate that K⁺ binding has an effect on the structural dynamics distinct from Na⁺. With a series of solid experiments in proteoliposomes, the authors demonstrate that K⁺ counter-transport facilitates sodium coupled dopamine uptake, uncovering a new aspect of transport energetics. The article is well-layered, starting from ensemble binding assays, then having a look at the structural effects with mass spectrometry and delving into the fine molecular details with single-molecule fluorescent measurements of ion fluxes. It is an excellent piece of work. It would however benefit from some more context regarding the experimental design and a few more introductory sentences (questions and suggestions below).

- Why is sometimes Cs⁺, sometimes NMDG⁺ used as a “non-interfering” control ion?
- Three different ligands were used throughout the paper: nisoxetine, MFZ 2-12 and dopamine. What is the rationale behind these choices? Is there a reason not to use dopamine for all the assays? Why use different ligands for the binding assay on hDAT and dDAT?
- I assume that the experiments were performed on dDAT because human DAT cannot at this stage be purified by the authors but this should be stated explicitly.
- Could the authors elaborate in the introduction about the physiological relevance of K⁺ in neurotransmission, what is known about the cellular concentration, and affinities to NSS transporters?
- Why were the HDX-MS experiments not repeated on biological replicates?

Minor comments:

- Abstract: ...and GABA that that utilize the energy... - that is repeated
- Intro : line 13 – basal transport mechanism is misleading, line 15 – embraces might be an odd choice of word.
- Figs: On the HDX-MS figures 2 and 3, the differential uptake is plotted as a function of the peptide number reported in supp. Table 3. Could these numbers also be added to supplementary fig.2 that shows the uptake per peptide?

Reviewer #2 (Remarks to the Author):

Schmidt et al. studies support a provocative conclusion that ligand binding to drosophila DAT (dDAT) and human DAT is inhibited by K⁺, and the conformational dynamics of dDAT in K⁺ are divergent from the apo- and Na⁺-states. The data presented in this paper reevaluate the impact of K⁺ on dopamine transport. The data presented in this paper strongly support a correlation between the increase in Na⁺ transport and K⁺ counter-transport rates in the presence of dopamine. These data are consistent with the novel idea of coupling of K⁺ to dopamine transport and increased uptake rate. While this reviewer found the data presented in this paper highly novel, rigorous, and transformative, additional experiments and justifications are needed to strengthen the conclusions' validity.

The justification for studying dDAT vs. hDAT is unclear. Page 4, paragraph one states, "Dopamine affinity and transport rate by dDAT is comparable to human DAT (hDAT), and the pharmacological profile of dDAT lies between that of hDAT and the human NET..." suggest hDAT would have been a better candidate. Some of the early figures used hDAT, some dDAT. The distinction(s) was not clearly stated.

K⁺ induced changes in the conformational state of the reconstituted dDAT (or any membrane-embedded protein) in detergent-lipid micelles should be interpreted cautiously, because the lipid composition of liposomes can produce artifact(s).

Using reconstituted dDAT in liposomes (i.e., synthetic vesicles) and membrane harvested hDAT are justified and routine systems for proof-of-concept studies. But the authors should show either K⁺ regulation of the uptake or binding in one of their available live cell models (Dr. Loland has used many DAT-expression cell lines in the past).

Overall, this is an outstanding work providing a previously unknown perspective in developing drugs that modulate dopamine transporter activity.

Reviewer #3 (Remarks to the Author):

The authors report on the dopamine transporter DAT and, in particular, they visualized Na⁺ and K⁺ fluxes from single proteoliposomes using fluorescent ion indicators. Total Internal Reflection (TIRF) microscopy appears to be an appropriate method for this purpose. However, in this context the

following question arises: How can the authors select single liposomes for their evaluation, since they illuminate an area of $82 \times 82 \mu\text{m}^2$ diameter containing about 200–250 liposomes? If they select a certain number of pixels in their image covering the surface of a liposome, they should consider that these liposomes have almost spherical shape with different TIRF illumination across their surface (depending on the relevant distance from the substrate). This may have an influence on the data shown in Figure 5 and further evaluated in the kinetics described on page 17. To estimate a possible error, it would be helpful, if the authors could add a few data, in particular size of the liposomes, number of pixels for evaluation of each liposome, as well as the TIRF angle of illumination (to estimate changes of illumination intensity over the spherical shape of a liposome).

General remark: Delete “that” in the 3rd line of the abstract.

Response to reviewer's comments to our manuscript entitled "The dopamine transporter antiports potassium to increase the uptake of dopamine"

We thank all reviewers for taking their time to thoroughly review our manuscript. We find your comments very insightful. We find that the manuscript has greatly improved after incorporating all the suggestions provided by the reviewers. Please see our replies and the resulting changes in the manuscript text, inserted as point-by-point in blue below your comments-

Please note that additional minor changes also occurs in the manuscript. These are suggestions for changes provided by other peers as a result of posting the manuscript on Research Square. These additions clarifies existing points and do not influence the interpretation of our findings. In order not to confuse them with the requests from the reviewer's comments, we chose not to annotate them. We will be happy to provide, if requested.

Sincerely,

Claus J. Loland

REVIEWER COMMENTS

Reviewer #1 (Remarks to the Author):

In this article, Schmidt and colleagues investigate the role of K⁺ ion on the transport mechanism of the NSS neurotransmitter dDAT. They first show that K⁺ competes with Na⁺ for binding to dDAT, then use HDX-MS to illustrate that K⁺ binding has an effect on the structural dynamics distinct from Na⁺. With a series of solid experiments in proteoliposomes, the authors demonstrate that K⁺ counter-transport facilitates sodium coupled dopamine uptake, uncovering a new aspect of transport energetics. The article is well-layered, starting from ensemble binding assays, then having a look at the structural effects with mass spectrometry and delving into the fine molecular details with single-molecule fluorescent measurements of ion fluxes. It is an excellent piece of work. It would however benefit from some more context regarding the experimental design and a few more introductory sentences (questions and suggestions below).

We thank the reviewer for the compliments.

- Why is sometimes Cs⁺, sometimes NMDG⁺ used as a "non-interfering" control ion?

Reply: NMDG⁺ has traditionally been used as a non-interfering or inert ion in electrophysiological studies, also regarding DAT and SERT (e.g. Carvelli et al., 2004 PNAS; Erreger et al., 2008 J Neurosci; Lin et al., 1996 Biophys J; Schicker et al., 2012 JBC; Coleman et al., 2019 Nature). In the HDX-MS experiments, we chose Cs⁺ because our previous HDX-MS experiments exploring the impact of Na⁺ and dopamine on dDAT were done using Cs⁺ as reference cation (see ref 20: Nielsen et al., 2019 Nat Commun). This way our results can be compared with the prior study. In the uptake studies with dDAT inserted in PLs, we did parallel experiments with both ions to show they are indifferent substitutes for K⁺ (fig 4b).

To emphasize this reasoning, we changed the text to (line 127):

"We compared the HDX data of dDAT in the presence of 200 mM K⁺ relative to a control buffer of similar ionic strength where K⁺ was substituted with Cs⁺. Cs⁺ was chosen as inert ion over NMDG⁺ because it is more K⁺ and Na⁺-like in size and because it enables comparison to previous HDX-MS studies of dDAT"

- Three different ligands were used throughout the paper: nisoxetine, MFZ 2-12 and dopamine. What is the rationale behind these choices? Is there a reason not to use dopamine for all the assays? Why use different ligands for the binding assay on hDAT and dDAT?

Reply: We do understand that this can be confusing. Due to the omission of a washing step, scintillation proximity assay opens for the possibility to assess binding of low-affinity radioligands. Unfortunately, the required copper-chelated scintillation beads used to immobilize the purified dDAT binds [3H]dopamine (and all other monoamines) even in the absence of dDAT. This results in an immense background which completely overwhelms the specific signal. The pharmacological profile of dDAT lies between that of hDAT and the human NET (see line 90) making [3H]nisoxetine the most suitable radioligand. For the hDAT binding experiments on membranes, [3H]dopamine is a ligand with too low affinity. Here the specifically bound [3H]dopamine will dissociate during the required washing step. The high-affinity cocaine analogue [3H]MFZ 2-12 is the most suitable radioligand for performing hDAT binding assays (see e.g. Newman et al., 2001 Bioorg Med Chem Lett). [3H]dopamine can only be used for transport assays where the radioligand is sequestered into the PLs or intact cells.

To clarify why the various ligands have been used, the text has been changed in two places:

(Line 93) “The potent NET inhibitor nisoxetine binds dDAT with high affinity (Supplementary Fig. 1a)²⁵. We investigated if there are indications that K⁺ can bind to dDAT by observing how [3H]nisoxetine binding was influenced by K⁺. In SERT and LeuT the interaction with K⁺ excludes substrate binding and is competitive to Na⁺ ^{26,19}. If a K⁺ binding site exists in dDAT, we expected it to have similar mechanistic features. To address this, we expressed full-length dDAT-8-His protein in suspension HEK293 cells (Expi293F) using the BacMam system^{27,28} and purified the transporter²³. We probed the Na⁺-dependence of [3H]nisoxetine binding to dDAT in mixed detergent-lipid micelles by the scintillation proximity assay²⁹ (Fig. 1a).”

(Line 108): “We expressed hDAT in Expi293F cells, harvested the membranes, and measured the Na⁺-dependence of binding of the high affinity cocaine analog [3H]MFZ 2-12² to hDAT in the presence and absence of K⁺ (Fig. 1c)”.

- I assume that the experiments were performed on dDAT because human DAT cannot at this stage be purified by the authors but this should be stated explicitly.

Reply: The reviewer is correct, and we agree that this was not explicitly stated. The text has been changed accordingly.

(Line 86) “ To address this aspect of the NSS mechanism, we turned to the *Drosophila* DAT (dDAT) which shares more than 50% sequence identity with the mammalian dopamine transporters^{23,24}. At this stage, it has not been possible to purify the human DAT (hDAT) in sufficient quantities and stability needed for the required experiments. However, dopamine affinity and transport rate by dDAT is comparable to hDAT, and the pharmacological profile of dDAT lies between that of hDAT and the human NET²⁵, making dDAT a suitable model for studies of NSS transport. Using dDAT, we investigated whether K⁺ is a component in dopamine transport.”

When the experiments are performed on hDAT it is now explicitly stated as a headline to the figure panel, see Fig. 1c and Fig 4e+f.

- Could the authors elaborate in the introduction about the physiological relevance of K⁺ in neurotransmission, what is known about the cellular concentration, and affinities to NSS transporters?

Reply: Yes, we agree with this suggestion. The text in the introduction has been expanded, by insertion of the following:

Line 66: "In the brain, SERT, DAT and NET are localized extrasynaptically in monoaminergic neurons¹⁵, which entails that during resting potential the transporters will be exposed to both an inward directed Na⁺ gradient and an outward directed K⁺ gradient. It has been long known that the serotonin transporter (SERT) in addition to the symport of Na⁺ catalyzes antiport of K⁺.¹⁶ The K⁺ antiport is permissive and increases the k_{cat} for serotonin¹⁷, but the affinity for K⁺ has not been determined. This feature is thought to be unique for SERT among the NSSs¹⁸. However, we have observed an effect of K⁺ on the transport properties and conformational state of LeuT, a prokaryote NSS¹⁹⁻²¹, with an apparent affinity for K⁺ to LeuT around 175 mM"

- Why were the HDX-MS experiments not repeated on biological replicates?

Reply: The HDX-MS experiments were performed as three technical replicate measurements (n = 3) for each time point. This is in accordance with the published recommendations for performing HDX-MS experiments (see ref #50 Masson et al., 2019 Nat Methods). We have qualified the HDX data against a preliminary dataset obtained from a previously purified batch of dDAT. We have specified and added this to the method section:

(Line 385) "All time points were performed as recommended⁵⁰ in three technical replicate measurements (n = 3)."

-and added (line 423): "The HDX data presented here were qualified against a preliminary dataset obtained from a previously purified batch of dDAT."

Minor comments:

- Abstract: ...and GABA that that utilize the energy... - that is repeated

Reply: This has been corrected.

- Intro : line 13 – basal transport mechanism is misleading,

Reply: Thank you, we agree. (line 58) "Basal" has been changed to "basic".

line 15 – embraces might be an odd choice of word.

Reply: The text has been changed from "It embraces..." to (Line 60) "NSSs are found in...".

- Figs: On the HDX-MS figures 2 and 3, the differential uptake is plotted as a function of the peptide number reported in supp. Table 3. Could these numbers also be added to supplementary fig.2 that shows the uptake per peptide?

Reply: The peptide numbers have now been added to the uptake plots in supplementary fig. 2.

Reviewer #2 (Remarks to the Author):

Schmidt et al. studies support a provocative conclusion that ligand binding to drosophila DAT (dDAT) and human DAT is inhibited by K⁺, and the conformational dynamics of dDAT in K⁺ are divergent from the apo- and Na⁺-states. The data presented in this paper reevaluate the impact of K⁺ on dopamine transport. The data presented in this paper strongly support a correlation between the increase in Na⁺ transport and K⁺ counter-transport rates in the presence of dopamine. These

data are consistent with the novel idea of coupling of K⁺ to dopamine transport and increased uptake rate. While this reviewer found the data presented in this paper highly novel, rigorous, and transformative, additional experiments and justifications are needed to strengthen the conclusions' validity.

We thank the reviewer for the positive comments and outlying the novelty of our findings.

The justification for studying dDAT vs. hDAT is unclear. Page 4, paragraph one states, "Dopamine affinity and transport rate by dDAT is comparable to human DAT (hDAT), and the pharmacological profile of dDAT lies between that of hDAT and the human NET..." suggest hDAT would have been a better candidate. Some of the early figures used hDAT, some dDAT. The distinction(s) was not clearly stated.

Reply: This is an important point also raised by Reviewer #1. To accommodate these comments from reviewer #1 and #2, we have further elaborated on the choice of transporter in the first paragraph of the Results section. It now reads:

(Line 86) To address this aspect of the NSS mechanism, we turned to the *Drosophila* DAT (dDAT) which shares more than 50% sequence identity with the mammalian dopamine transporters^{23,24}. At this stage, it has not been possible to purify the human DAT (hDAT) in sufficient quantities and stability needed for the required experiments. However, dopamine affinity and transport rate by dDAT is comparable to hDAT, and the pharmacological profile of dDAT lies between that of hDAT and the human NET²⁵, making dDAT a suitable model for studies of NSS transport. Using dDAT, we investigated whether K⁺ is a component in dopamine transport."

All subheadings and figures legend titles state when dDAT is investigated to not cause confusion with hDAT. In addition, when the experiments are performed on hDAT it is now explicitly stated as a headline to the figure panel, see Fig. 1c and Fig 4e+f.

K⁺ induced changes in the conformational state of the reconstituted dDAT (or any membrane-embedded protein) in detergent-lipid micelles should be interpreted cautiously, because the lipid composition of liposomes can produce artifact(s).

Reply: We agree that the lipid composition in both detergent-lipid micelles and liposomes can affect the conformational state and function of dDAT (and other membrane-embedded proteins). In our studies, we have kept the lipid composition of the detergent-lipid micelles or the liposomes constant and varied the ion composition of the buffers. For this reason, and because our protein is functional, we interpret the differences we observe to be attributed to the influence of the ions. However, if the lipid composition was changed it seems likely that the dDAT would behave differently in the assays and the absolute values measured would be different. To accommodate this comment, we have included the following in the discussion:

Line 277: "The properties of dDAT are expected to be highly sensitive to the lipid composition of the detergent micelles and the liposomes not least because of the cholesterol binding sites observed in the transporter^{24,35}, thus it is possible that the magnitude of the effect of the ions would be different in a different lipid composition."

Using reconstituted dDAT in liposomes (i.e., synthetic vesicles) and membrane harvested hDAT are justified and routine systems for proof-of-concept studies. But the authors should show either K⁺ regulation of the uptake or binding in one of their available live cell models (Dr. Loland has used many DAT-expression cell lines in the past).

Reply: We thank the reviewer for this great suggestion. To accommodate the request, we have performed [³H]dopamine uptake experiment on COS-7 transiently transfected with hDAT or dDAT (See Fig. 4e-f). We then assessed the effect of increasing the extracellular K⁺ concentration, which according to our findings from dDAT in PLs should decrease the uptake of [³H]dopamine. We performed the experiment in an uptake buffer where we had decreased the NaCl concentration to 30 mM and increased the KCl concentration to 105 mM, thus maintaining the total ion concentration close to the usual level. As control, we performed the experiment substituting K⁺ with NMDG⁺. Interestingly, the results very much parallels our observations from dDAT in PLs (Fig 4d). We observed that dopamine uptake decreased with addition of extracellular KCl, thus dissipating the K⁺ gradient. Our interpretation is cautious as the extracellular K⁺ could influence e.g. channel activities etc. causing an indirect effect on the transport rate. However we did not see any effect on cell viability by the brief incubation in K⁺ during [³H]dopamine transport. We have added the data as Fig. 4e-f in the manuscript (see below).

We have also added the following paragraph to the Results section:

(Line 182) **Dopamine transport into cells is inhibited by K⁺ in the uptake buffer**

To investigate how K⁺ affects dopamine uptake in a more complex system, we expressed hDAT in COS-7 cells. We compared the uptake of [³H]dopamine over 7.5 min in COS-7 cells in a buffer with 30 mM Na⁺ and either 105 mM K⁺ or NMDG⁺. We observed a significant decrease in V_{max} and an unaltered K_m for [³H]dopamine uptake when applying the external K⁺ (Fig. 4e). Although the experiment could influence many cellular processes, the effect is similar to our findings for [³H]dopamine uptake in PLs containing dDAT and could be interpreted as an inhibitory effect by the abolishment of the K⁺ gradient. Similarly, we expressed dDAT in COS-7 cells and performed the same uptake experiments. Again, we observed a marked decrease in dopamine uptake, in the presence of extracellular K⁺ although the lower affinity for dopamine to dDAT did not allow for a precise estimate of V_{max} and K_m (Fig. 4f). Taken together the uptake from whole cells could indicate that a dissipation of the K⁺ gradient not only affects transport in PLs but also in cells and the effect is also seen for hDAT.

- in the Discussion, we have added:

(Line 280) "An effect of K⁺ on uptake of dopamine does appear also to be present in COS-7 cells transiently transfected with both dDAT and hDAT, which represents a more native membrane environment for the transporters than liposomes."

- and changed this sentence to include observations from hDAT:

(Line 303) "The pharmacological data and the transport data on cells we present here on hDAT opens the possibility that K⁺ could have a similar effect in both drosophila and human dopamine transport. "

We added the following description of the new experiment in Methods:

(Line 512) "[³H]Dopamine uptake in intact COS-7 cells. COS-7 cells were transiently transfected using lipofectamine with 0.35 µg hDAT in pcDNA3.1 or 1 µg dDAT in pEG BacMam vector per 10⁶ cells in a lipofectamine:DNA ratio of 3.5. Cells were seeded in 24-well plates (5*10⁴ pr well) coated with poly-ornithine and incubated in 10% CO₂ at 37°C. [³H]dopamine transport assays were performed 48 hours after transfection. The transport experiments were carried out at room temperature. Prior to the experiment, each well were washed with 450 µl uptake buffer (30 mM NaCl, 25 mM HEPES, 1.2 mM CaCl₂, 1.2 mM MgSO₄, 1 mM ascorbic acid, 5 mM glucose, pH 7.4) supplemented with either 105 mM NMDG-Cl or 105 mM KCl. Uptake was performed in the indicated uptake buffer by addition of [³H]DA (0.6 Ci/mmol) [0.1; 12.5 µM] in a total volume of 500 µl. After 7.5 min the uptake reaction was stopped by two washes with 500 µl ice-cold uptake buffer. Cells were lysed in 1% SDS for 60 min at 37°C, then transferred to 24-well counting plates and counted in a 2450 MicroBeta² microplate counter (PerkinElmer, Waltham, MA) after addition of 500 µl Opti-phase Hi Safe 3 scintillation fluid. All experiments were performed in triplicates. Non-specific uptake was determined in cells pre-incubated with either 10 µM nomifensine (hDAT) or 100 µM nortriptyline (dDAT). Specific uptake data were fitted to the Michaelis-Menten equation in GraphPad Prism 7.0. "

Overall, this is an outstanding work providing a previously unknown perspective in developing drugs that modulate dopamine transporter activity.

Thank you very much.

Reviewer #3 (Remarks to the Author):

The authors report on the dopamine transporter DAT and, in particular, they visualized Na⁺ and K⁺ fluxes from single proteoliposomes using fluorescent ion indicators. Total Internal Reflection (TIRF) microscopy appears to be an appropriate method for this purpose. However, in this context the following question arises: How can the authors select single liposomes for their evaluation, since they illuminate an area of 82'82 µm² diameter containing about 200-250 liposomes? If they select a certain number of pixels in their image covering the surface of a liposome, they should consider that these liposomes have almost spherical shape with different TIRF illumination across their surface (depending on the relevant distance from the substrate). This may have an influence on the data shown in Figure 5 and further evaluated in the kinetics described on page 17. To estimate a possible error, it would be helpful, if the authors could add a few data, in particular size of the liposomes, number of pixels for evaluation of each liposome, as well as the TIRF angle of illumination (to estimate changes of illumination intensity over the spherical shape of a lysosome).

Reply: We are grateful for the reviewer's constructive comments and for giving us the chance to rectify some elements that deserve better explanation. We will first address the concerns regarding the liposomes density diameters and number of pixels and secondly TIRF illumination profile and penetration depth.

Liposome sizes. All liposomes imaged in these experiments are below, or close to, the optical resolution and thus appear as diffraction limited spots. To clarify this further we have attached a microcopy image clearly showing a high number (~230) of liposomes to appear as spherical dots, (as seen in Figure 1 below) that can be clearly defined and can be analyzed individually as we have applied and published extensively in the past¹⁻⁵. We have also attached here a zoom in, directly visualizing the signal extraction used for individual vesicles and local background correction. See below for changes in manuscript.

Number of pixels per liposome: To provide the number of pixels per liposome, we extract the intensity of each individual liposome using our previously published software^{1,4-7}. In short, the subpixel localization of all bright spots on a surface is found using a thresholded local maxima search and centroid fitting. Once a liposome has been localized, the signal is integrated as shown in Fig. 1, The green circle around the liposome displays the area used for local background correction, while the middle hole displays the area used for extraction of signal. The area outside the green circle is discarded for this particular liposome. In total, the background area is comprised of 88 pixels, while the integration area is 97 pixels. A large integration area ensures that even bright (large) vesicles will provide reliable signal. We note the values can be readily extracted from our software (available on hatzakislabsGitHub under MIT licence).

Figure 1 Signal extraction and background correction of diffraction limited proteoliposomes. A) Representative TIRF image displaying individual liposomes tethered to the glass surface. B) Zoom in on a single vesicle displaying the spot and the size of the signal extraction ROI (middle of the green circle, 97 pixels for all vesicles) and background ROI (green 88 pixels for all vesicles).

See below for changes in manuscript.

Liposome radius: We had originally provided the average hydrodynamic radius of the proteoliposomes using dynamic light scattering (see Supplementary Figure 4d) showing a mean hydrodynamic radius of ~181 nm. To address the comment of the referee, we now provide a size of individual liposomes by integrating the membrane signal from vesicles reconstituted both with and without fluorescent indicator (see figure below). This shows a clear lognormal distribution of membrane intensity consistent with our earlier findings^{1,3,7}. Combining the integrated intensity which is known to be proportional to the actual liposome size³ with the DLS measurements, we can convert the membrane signal of relative liposome size into distributions of actual liposome sizes (can be seen in the updated supplementary figure 4e - shown below. The previous supplementary figure 4e is now Supplementary Figure 6.).

Supplementary Figure 4e: Integrated membrane signal from 5939 liposomes with encapsulated indicator and 3016 without encapsulated indicator. The square root of the membrane integrated signal is proportional to the liposome size, and shows a lognormal distribution of sizes as expected. Using the mean liposome size evaluated by DLS, the membrane intensity can be converted to liposome sizes in nm. We find no significant difference in liposome sizes with and without encapsulated indicators.

TIRF penetration length and illumination profile: We are aware, as the reviewer correctly mentions, that TIRF illumination is not flat and that penetrations depth may influence spot shape. However, having demonstrated that all vesicles are in fact diffraction limited, we wish to highlight that the used penetration depth of 200 nm (corresponding to angles of 59.97, 60.23 and 60.78 degrees for 488 nm, 532 nm and 640 nm laser lines respectively) ensures full illumination of proteoliposomes. Additionally, the signal extraction displayed in Figure 1, ensures that the signal from each individual vesicles is corrected for local background, thus adjusting for any variations in the illumination profile.

To confirm for the current setup, the potentially uneven illumination profile across the field of view does alter the intensity quantification and thus liposome sizes or kinetics extraction we provide in the figure below (supplementary figure 8) an overview of vesicles size (found as described above) versus x and y location in our images. Here both visual inspection and a Pearson correlation coefficient confirm no difference in vesicles size as a function of surface position. See below for changes in manuscript.

Changes in response to reviewer's comments

We thank the reviewer for these insightful comments. Having realized that the diffraction limited nature of the experiments was not directly accessible; we have updated the results section in the main text. It now reads:

(Line 204) “The procedure maintains the spherical morphology and structural integrity of the liposomes^{36,37}. Comparison between DLS and TIRFm measurements reveals an expected log normal size distribution centered at ~181 nm with no change upon indicator encapsulation (Supplementary Figure 4d, e). Because PLs are below or close to the optical resolution, they will appear as diffraction limited spots. The 200 nm penetration depth ensures full illumination and unbiased analysis of the PLs as we have shown in the past³⁹⁻⁴³. Additionally, uptake of [³H]dopamine was maintained (Supplementary Fig. 5a, b).“

In addition, we updated Supplementary Fig. 4e to show individual liposome sizes from TIRF experiments, confirming DLS by showing no difference in size between loaded and unloaded liposomes.

We have also updated the methods section to state explicitly the size of the signal integration roi by adding:

(Line 578) “To provide local background corrected signal, a circular area (97 pixels) is used to integrate the signal from each individual liposome, where an outer annulus (88 pixels) form the basis for background evaluation.”

And changed the methods section, accordingly. It now reads:

(Line 505) “Using an adapted version of previously published software³⁷, the background corrected signal from both the membrane dye and ion indicator were extracted to ensure correct signal integration and adjusting potential bias from the known uneven TIRFm illumination profile (Supplementary Fig. 8a, b)”

To show laser angles and discuss the advent of local background correction.

Furthermore, we have added Supplementary Fig. 8 (see below) to improve transparency.

Supplementary Figure 8. Background corrected liposome-membrane intensity assessed for bias in position in field of view. Background corrected liposome-membrane intensity for the entire y positions (a) and x positions (b) over the field of view from the TIRF microscope. Using the local background correction liposome intensity and evaluation is unaffected by the position in the field of view.

General remark: Delete “that” in the 3rd line of the abstract.

Reply: “that” has been deleted.

List of references within the responses to the reviewers:

1. Thomsen, R.P. *et al.* A large size-selective DNA nanopore with sensing applications. *Nature Communications*, doi.org/10.1038/s41467-019-13284-1 (2019)
2. Holmstrøm, T. *et al.* Carbohydrate-Derived Metal-Chelator-Triggered Lipids for Liposomal Drug Delivery. *Chemistry - A European Journal*, doi.org/10.1002/chem.202005332 (2021)
3. Hatzakis, N.S. *et al.* How curved membranes recruit amphipathic helices and protein anchoring motifs. *Nature Chemical Biology*, doi.org/10.1038/nchembio.213 (2009)
4. Malle, M. *et al.* Single particle combinatorial multiplexed liposome fusion mediated by DNA. Research Square, doi.org/10.21203/rs.3.rs-177728/v1 *Submitted Under Review*
5. Singh, P.D., Bohr S., Hatzakis, N. Direct Observation of Sophorolipid Micelle Docking in Model Membranes and Cells by Single Particle Studies Reveals Optimal Fusion Conditions. *Biomolecules*, doi.org/10.3390/biom10091291 (2020)
6. Stella, S. *et al.* Conformational Activation Promotes CRISPR-Cas12a Catalysis and Resetting of the Endonuclease Activity. *Cell*, doi.org/10.1016/j.cell.2018.10.045 (2018)
7. Bohr, S. *et al.* Label-Free Fluorescence Quantification of Hydrolytic Enzyme Activity on Native Substrates Reveals How Lipase Function Depends on Membrane Curvature. *Langmuir*, doi.org/10.1021/acs.langmuir.0c00787 (2020)

REVIEWERS' COMMENTS

Reviewer #1 (Remarks to the Author):

I am happy with the authors response to my and the other reviewers' comments, and recommend this paper for publication.

However, I'd like to insist on the necessity of performing measurements on biological replicates when working with a sample as prone to variability as a eukaryotic membrane protein in detergent micelles. In our lab, our experience is that there is a sample-to-sample variability due the purification procedure that can be quite important. Membrane proteins tend to co-purify with associated lipids and from one sample to another, the amount and identity of those can change. We have observed and quantified these changes in lipid:protein ratio with ATR-FTIR on different membrane proteins. Substrate binding and "crystallizability" were modulated by the lipids present. Unfortunately, this was never published, although this article reminds me that it should be.

In the present case, it is unlikely that repeating HDX-MS experiments will change the authors' overall conclusions. However, I would like to see the preliminary data mentioned by the authors included in the supplemental. Also, can they elaborate on what they mean by "the HDX data were qualified against a preliminary dataset obtained from a previously purified batch of dDAT"? Can they explain which criteria they use to ensure that these new measurements were comparable to the previous ones? I'm guessing they did 2 HDX-MS with and without Cs⁺ (or another ion) and saw a similar trend in 2 HDX values?

Not doing biological replicates is a bad practice in the HDX-MS field and we should all work toward changing that. The HDX-MS white paper (Masson et al) cited by the authors actually states "Biological replicates of the experiment should be conducted where possible. This would require additional preparations of the protein. The repeats ensure that the variability in exchange measurements that can be ascribed to post-translational modifications/differences in protein expression/purification or variable stoichiometry in reconstituted protein complexes is quantified. Biological replicates are especially important for proteins that require extensive sample preparation before HDX-MS".

Reviewer #2 (Remarks to the Author):

The Authors have nicely revised the manuscript and addressed this reviewer's concerns. The authors conducted additional experiments such as [³H]dopamine uptake analyses on COS-7 transiently transfected with hDAT or dDAT (Fig. 4e+f). These data have significantly improved the overall conclusions. Overall the data presented in the revised manuscript are highly novel and transformative.

Reviewer #3 (Remarks to the Author):

The authors gave very detailed and satisfactory explanations to my questions and modified their manuscript correspondingly. I was only missing one sentence explaining that due to the size of the liposomes around 200 nm and the penetration depth of the evanescent electromagnetic field around 200 nm, whole liposomes were illuminated, but not with the same intensity in all parts of their volume. However, since the authors nicely showed that the liposomes are all spherical and very similar in size, this probably does not play a major role concerning their results and their interesting findings.

Reviewer #1 (Remarks to the Author):

I am happy with the authors response to my and the other reviewers' comments, and recommend this paper for publication.

However, I'd like to insist on the necessity of performing measurements on biological replicates when working with a sample as prone to variability as a eukaryotic membrane protein in detergent micelles. In our lab, our experience is that there is a sample-to-sample variability due the purification procedure that can be quite important. Membrane proteins tend to co-purify with associated lipids and from one sample to another, the amount and identity of those can change. We have observed and quantified these changes in lipid:protein ratio with ATR-FTIR on different membrane proteins. Substrate binding and "crystallizability" were modulated by the lipids present. Unfortunately, this was never published, although this article reminds me that it should be. In the present case, it is unlikely that repeating HDx-MS experiments will change the authors' overall conclusions. However, I would like to see the preliminary data mentioned by the authors included in the supplemental. Also, can they elaborate on what they mean by "the HDX data were qualified against a preliminary dataset obtained from a previously purified batch of dDAT"? Can they explain which criteria they use to ensure that these new measurements were comparable to the previous ones? I'm guessing they did Δ HDX-MS with and without Cs⁺ (or another ion) and saw a similar trend in Δ HDX values?

Not doing biological replicates is a bad practice in the HDX-MS field and we should all work toward changing that. The HDX-MS white paper (Masson et al) cited by the authors actually states "Biological replicates of the experiment should be conducted where possible. This would require additional preparations of the protein. The repeats ensure that the variability in exchange measurements that can be ascribed to post-translational modifications/differences in protein expression/purification or variable stoichiometry in reconstituted protein complexes is quantified. Biological replicates are especially important for proteins that require extensive sample preparation before HDX-MS".

We agree with all the points that reviewer #1 put forward in this response. We agree that the current sentence is insufficiently elaborate and we appreciate this suggestion to clarify in further detail. We have accordingly added the exploratory dataset as a Supplementary Fig. 8 (see below). When comparing this uptake plot to the uptake plot in Supplementary Figure 2 it is evident, now also to the reader, that there is a good qualitative agreement between the two sets. Due to minor experimental optimizations to increase sequence coverage and signal (e.g. change from C18 to C8 columns and addition of ion mobility) we are not able to include the first data set into the main data set obtained here.

To elaborate on how we qualified the second data set against the first, we have expanded the statement in the Methods section. It now reads (line 466):

The HDX data presented were qualified against an exploratory dataset obtained from a dDAT sample expressed and purified from a separate batch of Expi293F cells. This HDX-MS experiment was preliminary in nature and served to explore the HDX of dDAT with or without Cs⁺ and K⁺ and Na⁺. The second HDX-MS experiment was slightly optimized based on the exploratory experiment and was more comprehensive and included more technical replicate measurements. Overall, there was a good qualitative agreement between the two HDX-MS experiments and similar trends were observed pertaining to the impact of Cs⁺ and K⁺ and Na⁺ on dDAT. Due to minor differences in the experimental details between the experiments, a direct quantitative comparison should not be performed. We have included the data from the exploratory HDX-MS experiment in Supplementary

Fig. 8 for comparison. All results included in this paper as well as the statistical analysis of error is based on data from the second experiment.”

We agree with the reviewer that in general a biological replicate measurement should be conducted where possible. We also agree that in general eukaryote membrane proteins can be prone to sample-to-sample variability between purifications. We also agree with the reviewer that in the present case, it would not change the overall conclusions. In the case of dDAT, which we have expressed and purified overall >10 times to conduct the experiments presented in this study, we do not observe sample-to-sample variability in terms of affinity for ligands or ions, differences in glycosylation pattern on SDS-PAGE gels or differences in elution profiles from size-exclusion chromatograms. For this reason, we consider our samples from protein purifications of this particular protein homogenous. We also have earlier experience with HDX-MS experiments on dDAT (Nielsen et al. 2019 Nat Commun) where we did not observe any noticeable variability either. This is not to argue against the practice to use biological replicates; we just put forward our reasoning underlying the decision of performing the second HDX-MS data set in technical replicates.

Supplementary Figure 8. Deuterium uptake plots for dDAT peptides in preliminary HDX-MS study. Deuterium uptake plots for representative peptides of dDAT showing the relative deuterium uptake as a function of labeling time (0.25 – 480 min) for the K⁺ (purple), Cs⁺ (black), and Na⁺-bound (orange) states. In the preliminary HDX-MS study, time points were done in singlicate (n = 1) for all states except for the 0.25 min time point of the Cs⁺ state, which was done in triplicate (n = 3). Maximum-labeled control samples are shown as grey circles at 1440 min (n = 4). Values are plotted as means with SEM values plotted as error bars for time points where n > 1. Source data are provided as a Source Data file.